# Novel Macromolecular and Biobased Flame Retardants Based on Cellulose Esters and Phosphorylated Sugar Alcohols

**DOI:** 10.3390/polym15153195

**Published:** 2023-07-27

**Authors:** Matay Kaplan, Michael Ciesielski, Sabine Fuchs, Christoffer Getterle, Frank Schönberger, Rudolf Pfaendner

**Affiliations:** 1Fraunhofer Institute for Structural Durability and System Reliability LBF, 64289 Darmstadt, Germany; matay.kaplan@lbf.fraunhofer.de (M.K.); frank.schoenberger@lbf.fraunhofer.de (F.S.); rudolf.pfaendner@lbf.fraunhofer.de (R.P.); 2Hamm-Lippstadt University of Applied Sciences, 59063 Hamm, Germany; sabine.fuchs@hshl.de (S.F.); christoffer.getterle@hshl.de (C.G.)

**Keywords:** flame retardant, biobased, sugar alcohol, acrylic anhydride, cellulose acrylate, esterification, Phospha-Michael addition, polypropylene-polyethylene copolymer

## Abstract

The increasing demand to provide sustainably produced plastic materials requires, a.o., the development of biobased flame retardants (FRs) for applications where flame retardancy is essential. To meet those challenging new sustainability requirements, a set of novel phosphorus-containing cellulose esters were synthesized by an efficient two-step procedure. In the first step, cellulose was treated with acrylic anhydride to synthesize acrylate-functionalized cellulose esters—more specifically, cellulose acrylate butyrate (CeAcBu) and propionate (CeAcPr). Subsequently, phosphorylated anhydro erythritol (PAHE), synthesized from the sugar alcohol erythritol, was added to the acrylate-functionalized cellulose esters via Phospha-Michael addition. For comparison a cellulose ester based on 6*H*-Dibenzo[*c*,*e*][1,2]oxaphosphorin-6-on (DOPO) was prepared analogously. The acrylate-functionalized cellulose esters and novel FRs were characterized by NMR spectroscopy. TGA investigations of PAHE-functionalized CeAcBu revealed an onset temperature of decomposition (2% mass loss) of approx. 290 °C. The novel PAHE-based FR was incorporated into a polypropylene-polyethylene copolymer (PP-*co*-PE) together with poly-tert-butylphenol disulfide (PBDS) (8 wt.%/2 wt.%) as a synergist. The PP-PE samples achieved V2 classification in the UL 94 V test. In addition, specimens of a rapeseed oil-based polyamide containing PAHE-functionalized CeAcBu at 20 wt.% loading yielded a V2 rating with short burning times.

## 1. Introduction

The overall worldwide production volume of polymers is stated to be approximately 390 million tons per year [1]. In 2021, nearly 50% of the total annual consumption in Europe were polyolefins, including polyethylene and polypropylene [2]. As a matter of fact, polymers are mostly combustible materials with high energy content. Therefore, flame retardancy has become an important research area for many economic sectors such as textile, construction, or electronical/electro industries but more importantly to ensure safety, e.g., in buildings or means of transportation. Because most people die from smoke inhalation rather than exposure to the fire [3], the heat and smoke release or the spreading of the flame after ignition and during the burning process have to be taken into account [4,5,6,7]. There are many different types of FRs. Besides their flame-inhibiting mechanism (gas/condensed phase active, chemical, or physical effects), most of them can be assigned to minerals, halogenated, phosphorous-, nitrogen-, and silicon-containing compounds [5,8]. Especially the halogenated FRs are very efficient as radical scavengers in the gas phase, which inhibit further combustion. They are economical, easily processable, and have only minor effects on the mechanical properties of the polymer due to their high efficiency and good compatibility with the polymer matrix [6,9]. However, some of them have already been banned because large amounts of toxic and corrosive gases are released during combustion, especially in combination with toxic synergists like antimony(III)oxide [10,11,12]. Other halogenated FRs persist in the environment and are bioaccumulative and additionally toxic [13]. Phosphorus-based FRs, in contrast, can either act as radical scavengers in the gas phase or decompose to less problematic degradation products in the condensed phase such as (poly)phosphoric acid and derivates, which catalyze dehydration and carbonization to form a protective carbon layer [14].

In terms of environmental and health concerns, the global demand for sustainable flame-retardant polymers continues to grow because of increasing stringent environmental regulations for halogenated flame retardants [15]. Therefore, the opportunity arises for developing novel and environmentally friendly as well as biobased flame-retardant materials. However, there are two main challenges which need to be considered when using natural materials as raw materials for FR syntheses. The compatibility with (mostly) hydrophobic polymers is limited due to their hydrophobic nature. Secondly, the flammability and low degradation temperatures of natural materials limit their application as FRs. For instance, cellulose-based fibers typically release moisture above 100 °C and decompose somewhere between 200 and 340 °C [16].

Nevertheless, there are ways to overcome those difficulties by exploiting their strengths in other manners [6,17]. Besides the economic aspects (widespread, low cost, and easy access), they show fascinating complex natural structures, which are not easily adaptable by man-made synthetic products. The natural structures of chitosan, lignin, cyclodextrin, starch, and, most importantly for the current work, cellulose contain numerous hydroxyl groups in the polymeric backbone, which are accessible to numerous chemical reactions [18]. Their jointed carbonaceous backbone makes them unique as macromolecules and especially suitable as starting materials for FRs due to their ability to produce a thermally stable charred layer after exposure to fire. The macromolecular structure of cellulose, especially for high molecular weights, gives rise to several advantages for an application as a FR, because they cause fewer plasticizing effects and barely tend to migrate out of the material after compounding, which makes them superior compared to low molecular weight FRs [19,20]. Their chemical modification, e.g., with phosphorous groups, is necessary to render them active flame-retardant.

FRs are needed in different polymers, among them polyolefins. Polypropylene homo-/copolymers, for example, are widely used in pipes, automotive parts, electrical applications, films, and filament fibers [21]; although, pure polypropylene (PP) has one of the largest fire growth indices with an LOI (limiting oxygen index) of ~17.4% [22]. Hence, the use of adequate FRs is inevitable [23,24]. Polypropylene is typically flame- retarded with brominated FRs like decabromodiphenyl ethane, often in combination with antimony(III)oxide [21]. More recently, increased efforts to render polypropylene flame-retardant via halogen-free flame-retardant additives have been in progress. For example, the use of phosphorus-containing 3,9-dimethyl-3,9-dioxa-2,4,8,10-tetraoxa-3,9-diphosphaspiro-5,5-undecane, commercially available as Aflammit^®^ PCO 900 from Thor GmbH (Speyer, Germany), in combination with radical generators such as organic disulfides [25], sulfenamides [23], oxyimides [26], or sterically hindered amines [27] has been described. Even though those FR combinations can reach good flame-retardant performances in, e.g., UL 94 vertical test setups, migration of the typically low-molecular-weight flame-retardant additives out of the polymer compounds along with environmental/health concerns remains an important issue and is addressed in the current study.

Phosphorous-based FRs are still in the focus of current research. Ai et al. [14], for example, developed a FR based on melamine phenyl hypophosphite. Depending on the composition, the system shows a V-0 classification by UL-94 rating with an LOI of ~(29–31)% in PP. Nevertheless, a huge amount of about 30 wt.% FR is necessary, and the low-molecular-weight FR might undergo migration out of the polymeric matrix [20]. Such problems can be solved by incorporating the FR into a polymeric backbone. This is the case for the well-known FR ammonium polyphosphate (APP), where ammonium ions are ionically bond to a polyphosphate backbone [28]. Based on this, it seems that a more complex structure with a polymeric backbone might be beneficial, and this is where biopolymers come to shine. 

In the last decade, there have been many different approaches to develop sustainable FRs based on complex biopolymers. As an example, Gebke et al. [3] pursued [29] the modification of, e.g., starch with phosphate salts in molten urea to obtain a FR with a high phosphate content. The purified and unpurified FRs were incorporated in wood fibers, tested, and compared to the commercially available FR Kappaflam T4/729. Purification yielded a less inhibiting, less soluble, hence less applicable product. The unpurified product exhibited a better flame inhibition, probably due to aggregation of the phosphorous-/nitrogen-based reactants. Therefore, the latter product was used for further investigations in different biopolymers [30], even though the low-molecular-weight components might undergo migration out of the material, which is not desirable.

In terms of stability, cellulose is a more suitable candidate than starch. Although their building block is similar (α- and β-*D*-Glucose), cellulose exhibits a more stable structure due to a more easy assembling of the polymeric chains. This results in very strong hydrogen bonding and dipolar interactions between those chains, which is the main reason for its insolubility in water and its particularly stable and partially crystalline structure [31,32]. In addition to its low price and biodegradability, it comes along with a naturally developed functionality, flexibility, and high strength by taking advantage of a hierarchical structure from nano-scale to macroscopic dimensions [32,33]. Despite all those advantages, cellulose and other biopolymers have disadvantages as well, which can be overcome by simple chemical modification of its naturally occurring hydroxyl groups. Most of the attempts, such as the promising approach published by Sun et al. [34], currently focus on the surface functionalization of cellulose, because the solubility of cellulose is still a tremendous challenge. Therefore, cellulose fibers were coated with an ammonium salt of *N,N*-dimethylene piperazine (methylphosphonic acid). Indeed, the coated fibers showed an increased flame retardancy but were not very efficient as a FR nor did they have the desired high biobased content. Therefore, the goal of the current study was to increase the solubility of cellulose by a unique activation method, while introducing fully biobased flame inhibiting groups based on sugar alcohols in a homogenous way.

Another recent approach for cellulose modification is the work of Chen et al. [35]. The research group developed cellulose acrylates, which were functionalized with 6*H*-Dibenzo[*c,e*][1,2]oxaphosphorin-6-on (DOPO) resulting in transparent films. Those films self-extinguished immediately after removal of the flame. The authors point out the versatile properties of such modified cellulose acrylate derivatives and their potential applications. Nevertheless, their approach does not include a sustainable or (atom-)economic synthesis route appropriate for an industrially feasible upscaling. Speaking of sustainability, the authors do not address the use of DOPO as a flame-inhibiting group. DOPO is typically synthesized from 2-phenylphenol and phosphorus trichloride [36] and, therefore, neither provides access to a biobased flame-retardant solution nor to a completely halogen-free one. As part of the current work, a novel biobased phosphacycle based on the sugar alcohol erythritol was successfully developed as a sustainable alternative to DOPO with the same efficiency. Erythritol is mostly produced by microbial conversion of low-molecular-weight carbohydrates using osmophilic fungi [37,38]. Although Chen et al. [35] evaluated the flame retardancy of the obtained films as coatings, the efficiency as a flame-retardant additive in a polymeric matrix was not assessed.

In addition to polyolefins, polyamides require non-migrating, halogen-free, and, in the best case, biobased flame-retardant solutions. In the present study, the rapeseed oil-based Ultramid^®^ Flex F29 (BASF SE, Ludwigshafen am Rhein, Germany) was used as a model polymer for polyamides. Glass-fiber-reinforced polyamide 6.6, which is used in automotive as well as E&E parts [37,38], is halogen-free through aluminum diethyl phosphinate (DEPAL), alone or in combination with melamine polyphosphate (MPP) [36,39].

For the biobased Ultramid^®^ Flex F29, the use of DEPAL is not an appropriate choice as it would reduce its transparency. In contrast, biobased cellulose ester FRs reported in the current study can be tailor-made to realize perfect miscibility with the polymer and maintain its transparency in the final application.

Herein, we report on an efficient method to synthesize acrylate-functionalized cellulose esters, their conversion to a novel biobased phosphorus-containing FR, and first promising results on their thermal and flame-retardant properties. The synthesis of the novel FR was performed with a simple yet versatile chemistry, with a gentle and sustainable approach of cellulose esterification and subsequent Phospha-Michael addition of a phosphorus-containing sugar alcohol derivative on the acrylate groups of the cellulose (see Figure 1). The esterification procedure (I) does not include the use of toxic solvents/reagents (e.g., acid chlorides, organic solvents/ionic liquids) or laborious purification methods [35,40,41,42]. The Phospha-Michael addition (II) was performed with phosphorylated anhydro erythritol (PAHE) as the Michael donor. PAHE is a phosphorus-containing bicyclic derivative of erythritol bearing a reactive P-H functionality and is easily accessible from the sugar alcohol by a simple literature-reported procedure [43,44]. In addition, a FR functionalized with the established 6*H*-Dibenzo[*c,e*][1,2]oxaphosphorin-6-on (DOPO) was analogously synthesized by Phospha-Michael addition as a reference system.

The novel and profoundly characterized FRs were incorporated into a polypropylene-polyethylene copolymer (PP-*co*-PE) alone and in combination with the synergist poly-*tert*-butylphenol disulfide (PBDS). Moreover, specimens of the rapeseed oil-based polyamide Ultramid^®^ Flex F29 containing the novel phosphorus-containing cellulose esters were prepared. The flame-retardant properties were evaluated by UL 94 vertical burning test. It is noteworthy to mention that the present study primarily focuses on the synthesis, structural characterization, and the thermal properties of the novel biobased additives. Therefore, only results of a preliminary assessment of the flame-retardant properties are currently provided.

Detailed investigations of the FRs’ effects and their mode of action as well as their structural optimization to extend their scope of application will be carried out in the frame of future studies.

## 2. Materials and Method

### 2.1. Materials

Both cellulose (100% bio-cotton) and the erythritol Xucker^®^ were purchased from a drugstore (DM, ebelin, Karlsruhe). Acrylic acid (99% anhydrous, 1 L, 200 ppm MEHQ), dimethyl phosphite (98%, 500 g), and triethylamine (99.5%, 500 mL) were received from Sigma-Aldrich (Taufkirchen, Germany). Acetic acid anhydride (99.6%, 2.5 L), zinc(II)chloride (98.3%, 500 g), sulfuric acid (95%, 2.5 L), and acetonitrile (99.95% anhydrous, 1 L) were purchased from VWR International (Darmstadt, Germany). Butyric anhydride (97%, 1 L), cyclohexane (>99.5%, 2 L), phenothiazine (≥97%, 100 g), and 4-methoxyphenol (MEHQ, ≥99%, 250 g) were purchased from Merck KGaA (Darmstadt, Germany). Anhydrous toluene (99.85%, 2.5 L) was obtained from Acros Organics (Waltham, MA, USA). Methyl t-butyl ether (MTBE, 99.8%, 4 L) was bought from Scharlab, S.L.(Barcelona, Spain). Dowex^TM^ 50WX8 was used from Thermo Scientific Chemicals (Waltham, MA, USA). Polypropylene-polyethylene copolymer (PP-co-PE) Moplen^®^ RP320 M was purchased from LyondellBasell (Rotterdam, The Netherlands): MFR/230/2,16 (test temperature 230 °C, mass 2.16 kg); 8 g/10 min. Polyamide Ultramid^®^ Flex F29 was kindly provided by BASF SE (Ludwigshafen, Germany). MFR/230/10: 113 g/10 min. Exolit^®^ OP 1230 (DEPAL) was provided by Clariant Plastics & Coatings (Deutschland) GmbH (Hürth, Germany); decabromodiphenyl ethane (DBDE, FR-1410) was received from ICL-IP Europe B.V. (Amsterdam, The Netherlands). The antimony(III)oxide (ATO, abcr GmbH (Karlsruhe, Germany)) masterbatch (50 wt.% ATO in polypropylene) was prepared with PP homopolymer 100-GA04 (INEOS Olefins & Polymers Europe, Cologne, Germany; MFR/230/2,16: 4 g/10 min on a 12 mm co-rotating twin-screw extruder (ThreeTec, Seon, Switzerland). Vultac^®^ TB7 (PBDS) was kindly provided by Safic-Alcan (Bad Kreuznach, Germany).

### 2.2. Methods

Liquid nuclear magnetic resonance spectroscopy (NMR) measurements were performed on a BRUKER Avance 300 MHz (MA, USA) in deuterated chloroform as well as deuterated dimethyl sulfoxide. The corresponding spectra (^1^H, ^31^P, ^13^C) were evaluated with the software MestReNova V.14.2 from Mestrelab Research. Due to the low magnetic field force, the integratable ^13^C spectra were measured for about 60 h with a relaxation time of 20 s in combination with chrome(III)acetylacetonate as a relaxation agent. Due to the paramagnetism of the latter, the long relaxation times for ^13^C nuclei can be reduced because another relaxation path is available.

Fourier-transform infrared spectroscopy (FT-IR) was performed on a Nexus FT-IR 670/870 from Nicolet Instrument Corporation (WI, USA). The spectra were evaluated with the help of Silverstein et al. [45].

UV-VIS measurements were carried out on a Shimadzu UV-2600i UV-VIS spectrophotometer (Kyōto, Japan).

Size-exclusion chromatography (SEC) was performed on an Agilent Infinity 1100 (Agilent, Waldbronn, Germany) with chloroform as a mobile phase (1 mL/min), three SDV columns as a stationary phase, and a polystyrene calibration (PSS Polymer Standards Service GmbH, Mainz, Germany). The detection was carried out using a DRI detector (Infinity 1100 Agilent). The sample was dissolved in chloroform (2 mg/mL) and filtered with a 0.45 µm PTFE syringe filter prior to injection.

Thermogravimetric analysis (TGA) was performed using a Mettler Toledo TGA/DSC 1 STAR System (Gießen, Germany) under a N_2_ atmosphere and with a heating rate of 10 °C/min.

Elemental analysis (EA) was conducted with a Vario Micro Cube from Elementar (Langenselbold, Germany). An amount of 2.0 ± 0.1 mg of each sample was placed in tin capsules before flushing the pyrolysis chamber with oxygen for 80 s at 1050 °C. 

UL 94 V flammability tests of polymer compounds (UL 94 V standard, Underwriters Laboratories Inc., Northbrook, IL, USA), complying with DIN EN 60695-11-10 [46]) were performed in a HVUL2 Horizontal Vertical Flame Chamber from Atlas Material Testing Technology GmbH (Linsengericht-Altenhaßlau, Germany).

Phosphorus contents of the synthesized flame retardants were commissioned by Mikroanalytisches Laboratorium Kolbe (Oberhausen, Germany).

### 2.3. Synthesis of Flame Retardants

Synthesis procedures described in Section 2.3.3, Section 2.3.4, Section 2.3.5, Section 2.3.6, Section 2.3.7, Section 2.3.8, Section 2.3.9, Section 2.3.10. were carried out under a nitrogen atmosphere, whereby every fifteen minutes air was added through a syringe to regenerate MEHQ as the inhibitor. The specific reactions are shown in Figure 2 and Figure 3.

#### 2.3.1. Synthesis of the Acrylic Anhydride Solution (AA–AH) (**2**)

The acrylic acid anhydride solution (**2**) was prepared by a simple distillation via a 50 cm × 2 cm mirrored Vigreux column in a 2 L three-neck round flask. A total amount of 383 g acetic acid anhydride (**2.1**) (3.75 mol, 1 eq.), 613 g acrylic acid (**2.2**) (8.5 mol, 2.3 eq.), and 0.5 g phenothiazine were needed, whereby the acrylic acid was added portion-wise during the distillation (5 mol, 2 mol, 2 × 0.75 mol). The oil bath was set to 93 °C to obtain a temperature inside the flask of about 80 °C in the gas phase. For the first portion of acrylic acid anhydride (5 mol), the pressure was reduced step-wise to 102 mbar, where the distillation of acetic acid (with some acrylic acid) started. The pressure was reduced slowly to 94 mbar within 6.5 h. The temperature at the top of the column was about 45–52 °C. The second portion of acrylic acid (2 mol) was added, and the distillation was continued for another 6.5 h. The distillate was collected at 61 °C/93 mbar. The third portion of acrylic acid (0.75 mol) was added with another 100 mg of phenothiazine and the distillation was continued for 7.5 h at 90 mbar. The last portion of acrylic acid (0.75) and 100 mg of phenothiazine were then added and distilled at 79 mbar. The pressure was reduced slowly to 62 mbar during the distillation (7.5 h) to maintain a temperature of about 57 °C above the column, then, for five more hours, the distillation was continued at 52 mbar. Finally, the distillation was continued by reducing the pressure slowly to 27 mbar at (55–60) °C above the column. The relative amount of the one-sided product was observed by ^1^H NMR spectroscopy. After confirmation of an amount lower than 3%, the raw product was finally distilled at 18 mbar/70 °C and 13 mbar/65 °C. At the end, the oil bath was set to 105 °C. With a temperature inside the flask of 55 °C, the last fraction of the product was collected at 5 mbar/50 °C. An amount of 364 g of acrylic acid anhydride (**2**) (1.98 mol) was collected as a clear colorless liquid. Yield: 77%.

**^1^H NMR** (300 MHz, CDCl_3_, ppm): δ 6.51 (*dd*, 1H, *cis-3-H*, ^2^*J*_HH_ = 16.83 Hz, ^3^*J*_HH_ = 1.41 Hz), 6.18–6.01 (*m*, 2H, *trans-3-H*, *2-H*), 2.22 (s, 3H, *2′-H*).

#### 2.3.2. Synthesis of Tetrahydrofuro[3,4-d][1,3,2]dioxaphosphole 2-oxide (**5-PAHE**) (Phosphorylated Anhydro Erythritol, PAHE)

PAHE was synthesized in conformity with the literature [43,44]. In the first step, a 250 mL Schlenk flask was equipped with a Liebig condenser and filled with 50 g (0.41 mol, 1 eq.) erythritol (**5.1**) and 2 g Dowex^TM^ 50WX8 under a nitrogen atmosphere. The oil bath was set to 140 °C and the mixture was stirred for 30 min. Then, the pressure was slowly reduced to 0.08 mbar and the oil bath temperature was raised to 180 °C. The solution started to boil, and the condensate was collected. The distillation finished after 45 min and led to a colorless viscous liquid. The raw product was purified again by distillation at 0.074 mbar/110 °C, leading to 34.14 g of anhydro erythritol (**5.2**) (0.33 mol). Yield: 80.5%

The phosphorylation of 34.14 g anhydro erythritol (**5.2**) (328 mmol) was carried out by an equimolar implementation with 36.1 g dimethyl phosphite (328 mmol) in a 100 mL round flask. The round flask was equipped with a Liebig condenser and the oil bath was set to 140 °C. The distillation of methanol was first carried out at atmosphere pressure (40 °C/atm). Afterwards, the pressure was reduced to 0.094 mbar, the oil bath temperature was raised to 150 °C, and the residual methanol was distilled within 45 min. After cooling the mixture to room temperature, platelet-like crystals started to form. Purification was performed by recrystallization of the molten crystals in 100 mL toluene. The crystals were then filtered and washed two times with 100 mL cyclohexane. An amount of 41.38 g of the phosphorylated anhydro erythritol (**5-PAHE**) (276 mmol) was collected as colorless platelet-like hygroscopic crystals. Yield: 84.1% (total yield: 67.7%).

**^1^H NMR** (300 MHz, CDCl_3_, ppm): δ 8.58, 6.15 (*d*, 1H, *6-H*, ^1^*J*_PH_ = n.d.), 8.54–6.14 (*d*, 1H, *6-H**, ^1^*J*_PH_ = n.d.), 5.20–5.16 (*m*, 2H, *cis-3-H*, *cis 5-H*), 5.10–5.07 (*m*, 2H, *cis-3-H**, *cis-5-H**), 4.25–4.21 (*m*, 2H, *1-H**, *2-H**), 4.04–4.00 (*m*, 2H, *1-H*, *2-H*), 3.57–3.53 (*m*, 2H, *trans-3-H**, *trans-5-H**), 3.49–3.45 (*m*, 2H, *trans-3-H*, *trans-5-H*).

**^31^P NMR** (300 MHz, CDCl_3_, ppm): δ 25.11 (*s*, 1P, *1′-P*), 23.97 (*s*, 1P, *1-P*), 7.58 (*m*, 1P, *1″-P*).

**^1^H NMR** (300 MHz, CDCl_3_, ppm, **isolated isomer**): δ 7.32 (*d*, 1H, *6-H*, ^1^*J*_PH_ = 722.96 Hz), 5.20–5.16 (*m*, 2H, *cis-3-H*, *cis 5-H*), 4.04–4.00 (*m*, 2H, *1-H*, *2-H*), 3.51–3.47 (*m*, 2H, *trans-3-H*, *trans-5-H*).

**IR** ($\bar{\nu }$/cm^−1^): 2858 m (sp^3^-C), 2511 w (P-H), 1464 w, 1331 m, 1254 s (P=O), 1227 s (P=O), 1093 s (P-O-C), 1024 s (P-O-C), 903 s, 779 s, 712 m.

#### 2.3.3. Synthesis of Cellulose Acrylate Butyrate (**4-CeAcBu**)

A 2 L round flask equipped with a KPG stirrer was filled with 32 g of cellulose (**1**) (99 mmol cellobiose units with approximately 590 mmol OH groups, 1 eq.; equivalents refer to the cellulose hydroxyl groups (three per ring)). The oil bath was set to 50 °C and the flask was desiccated three times at 0.02 mbar. An amount of 250 mg MEHQ and 250 mg phenothiazine in 330 g acrylic acid (4.6 mol) were added and the mixture was stirred for five minutes. Then, a mixture of 173 g of the acrylic acid anhydride solution (**2**) (1.37 mol, 2.3 eq.) from Section 2.3.1 and 11.2 g butyric anhydride (**3**) (70.8 mmol, 0.1 eq.) were added and stirred for ten more minutes. Finally, 6.9 g of anhydrous ZnCl_2_ (0.05 mol, 0.08 eq.) was added at 55 °C and stirred for five more minutes. Afterwards, the oil bath was set to 77 °C under severe stirring. After 45 min, another 6.8 g of ZnCl_2_ (0.05 mol, 0.08 eq.) with 150 mg MEHQ was added. At that point, the mush mixture started to liquefy. After 10 min, the temperature was raised to 82 °C for 1 h, then 87–90 °C for another 120 min. After homogenization (slight cloudy pale beige), the viscous solution was stirred for 30 min. Finally, the solution was cooled down to 75 °C and 1.1 L cyclohexane was added under stirring. Regarding the degree and type of esterification, the phase separation in this step was good. The granulate-like solid was easily filtered through an M-size glass frit, purified by stirring it six times with MTBE (+100 mg MEHQ) (700 mL, 650 mL, 600 mL, 550 mL, 2 × 500 mL) at 55 °C for about 7 min, cooled in a water–ice bath and filtered again over an M-size glass frit. The precipitate was carefully dried at 40 °C/10 mbar and yielded a white, fluffy, and easy-to-dust powder. Because spontaneous polymerization is possible, removal of all MTBE was not pursued further. An amount of 57.5 g of the cellulose acrylate butyrate (**4-CeAcBu**) was collected as a fine white powder, which can be stored in the fridge (−18 °C) for several months. Yield: 88%.

**^1^H NMR** (300 MHz, CDCl_3_, ppm): δ 6.49–5.65 (*m*, 3H, *1‴-H*, *2‴-H*), 5.12–4.90 (*m*, 2H, *1-H*, *2-H*), 4.35–4.01 (*m*, 3H, *3-H*, *4-H*, *5-H*), 3.69–3.44 (*m*, 2H, *6-H*), 2.31–2.09 (*m*, 2H, *1″-H*), 1.92–1.84 (*m*, 3H, *1′-H*), 1.67–1.49 (*m*, 2H, *2″-H*), 0.99–0.82 (*m*, 3H, *3″-H*).

**^13^C NMR** (300 MHz, CDCl_3_, ppm): δ 172.14 (*m*, 1C, *butyric carbonyl*), 169.54 (*m*, 1C, *acetic carbonyl*), 164.53 (*m*, 1C, *acrylic carbonyl*), 132.15 (*m*, 1C, *1‴-C*), 127.30 (*m*, 1C, *2‴-C*), 116.36 (*s*, *unknown*), 100.79 (*m*, 1C, *1-C*), 71.34 (*m*, Cellulose carbon, *not assignable*), 61.99 (*m*, 1C, *6-C*), 35.59 (*m*, 1C, *1″-C*), 20.62 (*m*, 1C, *1′-C*), 18.16 (*m*, 1C, *2″-C*), 13.60 (*m*, 1C, *3″-C*).

**IR** ($\bar{\nu }$/cm^−1^): 3356 w br (water), 2964 w/2879 w (sp^3^-C), 1724 s (C=O), 1633 w (C=C), 1406 m (acrylic C=O), 1257 m/1159 s/1051 s (ester C-O stretching), 980 m, 802 m.

**Molecular weight MW** (determined by size-exclusion chromatography): $\bar{M_{n}}=13$ kDa; $\bar{M_{w}}=73$ kDa; *Đ* = 5.6 (see Figure A5).

**Degree of esterification per ring** (determined by nuclear magnetic resonance spectroscopy): 2.1 acryl-, 0.85 butyl-, 0.25 acetyl groups.

#### 2.3.4. Synthesis of Phosphorylated Cellulose Acrylate Butyrate with PAHE (**6-CeAcBu-PAHE**)

A 500 mL round flask was sealed with a septum, desiccated, and flushed with nitrogen four times. An amount of 51 g PAHE (**5-PAHE**) (340 mmol, eq.) was added and dissolved in 225 mL anhydrous acetonitrile. Then, 500 mg of MEHQ and 55 g of the cellulose acrylate butyrate (**4-CeAcBu**) were dissolved separately in 350 mL anhydrous acetonitrile at 60 °C in a 2 L three-neck round flask equipped with a KPG stirrer, a nitrogen inlet, and a septum. After 10 min, 67 mL triethylamine (483 mmol, eq.) was added to the slightly cloudy solution. The PAHE solution was then added dropwise within 70 min, the temperature was set to 70 °C, and every 10 min, 20 mL of air was added through a syringe.

After another 30 min, the temperature was raised to 75 °C. After 60 min, 7 mL triethylamine was added at 77 °C. The progress of the reaction was observed by NMR spectroscopy. Generally, the reaction should be completed within 4 h after the first addition of the reactant. If there are still some free acryl groups available but all PAHE is used up, another 7 g PAHE should be added. The solution was cooled to 65 °C and the product precipitated in 550 mL anhydrous toluene. The product was collected as a white solid, which was purified by dissolution/dispersion in anhydrous acetonitrile (270 mL) at a 90 °C oil bath temperature followed by precipitation in anhydrous toluene (350–400 mL). The product was decanted at 37 °C. The collected white powder was finally dried carefully in vacuum (starting from 10 mbar at 60 °C) and ground afterwards. Finally, the collected powder was dried at 150 °C in high vacuum (1 × 10^−2^ mbar) and yielded 66.5 g of the desired cellulose ester FR as fine white powder (**6-CeAcBu-PAHE**). Yield: 65%.

**^1^H NMR** (300 MHz, d_6_-DMSO, ppm): δ 5.23–5.13 (*m*, 2H, *trans-6‴-H*, *trans-8‴-H*), 4.62 (*m*, 1H, *1-H*), 4.00 (*m*, 2H, *5‴-H*, *9‴-H*), 3.77 (*m*, Cellulose, *not assignable*), 3.49 (*m*, 2H, *cis-6‴-H*, *cis-8‴-H*), 3.31 (*m*, Cellulose, *not assignable*), 2.72–2.08 (*m*, overlap of: *1′-H*, *1″-H*, *2‴-H*, *3‴-H*), 1.58–1.46 (*m*, 2H, *2″-H*), 0.91–0.84 (*m*, 3H, *3″-H*).

**^13^C NMR** (300 MHz, CDCl_3_, ppm): δ 171.43 (*m*, 3C, *carbonyl*), 99.07 (*m*, 1C, *1-C*), 81.71–81.25 (*m*, 2C, *5‴-C*, *9‴-C*), 73.10–72.83 (*m*, 2C, *6‴-C*, *8‴-C*), 69.93 (*m*, Cellulose carbon, *not assignable*), 63.92 (*m*, 1C, *6-C*), 26.78 (*m*, 1C, *2″-C*), 21.36–17.53 (*m*, 4C, *1′-C*, *1″-C*, *2″-C*, *3‴-C*), 13.22 (*m*, 1C, *3″-C*).

**^31^P NMR** (300 MHz, d_6_-DMSO, ppm): δ 50.45 (*s*, 1P).

**IR** ($\bar{\nu }$/cm^−1^): 3359 w br (water), 2966 w/2933 w/2862 w (sp^3^-C), 1738 s (C=O), 1261 m/1026 s (ester C-O stretching), 918 m, 793 w, 704 w.

**Elemental analysis calc. (%): C:** 45.00, H: 5.35, P: 9.67

**found (%): C:** 43.65, H: 5.46, N: 0.04, P: 9.06.

#### 2.3.5. Synthesis of Phosphorylated Cellulose Acrylate Butyrate with DOPO (**6-CeAcBu-DOPO**)

A 1 L round flask equipped with a KPG stirrer, a nitrogen inlet, and a septum was desiccated and flushed with nitrogen four times. An amount of 150 mg phenothiazine and 18 g **4-CeAcBu** were dissolved in 200 mL anhydrous acetonitrile at an oil bath temperature of 70 °C in a nitrogen atmosphere, resulting in a colorless, viscous, and slightly cloudy solution. Then, 27 mL of dry triethylamine was added. A DOPO (**5-DOPO**) solution was prepared by dissolving 30 g of DOPO separately in 200 mL anhydrous acetonitrile under a nitrogen atmosphere. The DOPO solution was added within 45 min over a syringe and the temperature was raised to 80 °C. The solution became cloudier and was stirred for another 2.5 h. Then, the heat plate was turned off and stirring was continued for 15 min. The product precipitated and the filtrate was clear and colorless. The conversion of the reaction was observed successfully by NMR spectroscopy. The raw product was stirred in 125 mL anhydrous acetonitrile for 15 min at 72 °C vigorously, decanted, and stirred again in 140 mL anhydrous toluene at 90 °C. After filtration (G3 glass frit), the product was collected as a swollen solid, which was dried slowly under vacuum at 60 °C, ground, and dried at 90 °C in high vacuum (1 × 10^−2^ mbar). Because the product binds toluene strongly, drying and grinding need to be repeated four times. The drying process in total took over 8 h at 150 °C under high vacuum. The final NMR spectra (^1^H & ^31^P) are attached to the Supporting Information (Appendix A) and show a second peak in the ^31^P NMR, indicating a small degradation. This was most probably due to the severe drying process. Therefore, the synthesis needs to be optimized. Especially an alternative solvent for toluene needs to be found for the purification. The material can still be used as a reference, though.

**^1^H NMR** (300 MHz, CDCl_3_, ppm): δ 7.82–7.15 (*m*, 8H, *aromatic DOPO-H*), 4.96–4.72 (*m*, 2H, *1-H*, *2-H*), 4.30–3.59 (*m*, 5H, *3-H*, *4-H*, *5-H*, *6-H*), 2.49–2.22 (*m*, overlap of: *1′-H*, *1″-H*, *2‴-H*, *3‴-H*), 1.41 (*m*, 2H, *2″-H*), 0.76 (*m*, 3H, *3″-H*).

**^31^P NMR** (300 MHz, CDCl_3_, ppm): δ 35.84 (*s*, 1P, main product: *DOPO ring closed*), 37.06 (*s*, 1P, side product: *DOPO ring opened*).

**Elemental analysis calc. (%): C:** 62.18, H: 4.96, P: 8.02

**found (%): C:** 59.78, H: 4.93, N: 0.07, P: 7.41.

#### 2.3.6. Synthesis of Cellulose Acrylate Propionate (**4-CeAcPr**)

A desiccated 100 mL three-neck Schlenk round flask was equipped with a magnetic stirrer and 1.2 g of cellulose (**1**) (4 mmol cellobiose units with approximately 22 mmol OH groups, 1 eq.; equivalents refer to the cellulose hydroxyl groups (three per ring)). The flask was dried in vacuum (1 × 10^−2^ mbar) and filled with nitrogen. An amount of 12 g of acrylic acid was added with 10 mg of phenothiazine and the system was stirred for 3 min. An amount of 7 g of the home-made acrylic anhydride solution (55.5 mmol, 2.5 eq.; equivalents refer to the cellulose hydroxyl groups (three per ring)) was added with 0.36 g of propionic anhydride (2.8 mmol, 0.13 eq.; equivalents refer to the cellulose hydroxyl groups (three per ring)) and stirred for a couple of minutes. An amount of 0.25 g of ZnCl_2_ (1.8 mmol, 0.08 eq.; equivalents refer to the cellulose hydroxyl groups (three per ring)) was added and stirred at room temperature for ten minutes. The temperature was raised within 40 min to 57 °C and 10 min later to 63 °C. After 1 h, the temperature was again raised to 75 °C. After addition of another 0.3 g ZnCl_2_ (2.2 mmol, 0.1 eq.; equivalents refer to the cellulose hydroxyl groups (three per ring)), the temperature was raised to 85 °C and the mush started to liquify. Then, 1.5 h later, the solution became homogenous. After 2 h, the temperature was raised to 90 °C for 40 min.

Finally, the product precipitated at 50 °C after the addition of 80 mL cyclohexane. The solution was decanted, and the brown precipitate was stirred in 80 mL MTBE at 55 °C, cooled, and decanted again. This was repeated three more times and the product was dried under vacuum (1 × 10^−2^ mbar) at 55 °C. Yield: 80%.

**^1^H NMR** (300 MHz, CDCl_3_, ppm): δ 6.49–5.66 (*m*, 3H, *1‴-H*, *2‴-H*), 5.13–4.92 (*m*, 2H, *1-H*, *2-H*), 4.36–4.03 (*m*, 3H, *3-H*, *4-H*, *5-H*), 3.71–3.46 (*m*, 2H, *6-H*), 2.37–2.10 (*m*, 2H, *1″-H*), 1.96–1.86 (*m*, 3H, *1′-H*), 1.16–1.00 (*m*, 3H, *2″-H*).

**Degree of esterification per ring** (determined by nuclear magnetic resonance spectroscopy)**:** 2.2 acryl-, 1 butyl-, 0.2 acetyl groups.

#### 2.3.7. Synthesis of Cellulose Acrylate Propionate (**4-CeAcPr^SA^**)

A desiccated 500 mL three-neck Schlenk round flask was equipped with a mechanical stirrer and 15 g of cellulose (**1**) (46 mmol cellobiose units with approximately 277 mmol OH groups, 1 eq.; equivalents refer to the cellulose hydroxyl groups (three per ring)). The flask was dried in vacuum (1 × 10^−2^ mbar) and filled with nitrogen. An amount of 150 g of acrylic acid was added with 100 mg of phenothiazine and the system was stirred for 3 min. An amount of 87 g of the home-made acrylic anhydride solution (690 mmol, 2.5 eq.; equivalents refer to the cellulose hydroxyl groups (three per ring)) was added with 4.5 g of propionic anhydride (35 mmol, 0.13 eq.; equivalents refer to the cellulose hydroxyl groups (three per ring)) and stirred for a couple of minutes. An amount of 1.25 g of conc. H_2_SO_4_ was added and stirred at room temperature for ten minutes. The temperature was raised within 40 min to 57 °C. After 2 h, the mush started to liquify and the reaction was continued for another 2 h. The low boiling reagents of the resulting homogenous solution were distilled under vacuum at 55 °C.

Finally, the product precipitated at 55 °C after the addition of 670 mL cyclohexane. The solution was filtered, and the brown precipitate was stirred in 200 mL MTBE at 55 °C, cooled, and decanted again. This was repeated four more times and the product was dried under vacuum (1 × 10^−2^ mbar) at 60 °C. Yield: 76%.

**^1^H NMR** (300 MHz, CDCl_3_, ppm): δ 6.51–5.65 (*m*, 3H, *1‴-H*, *2‴-H*), 5.12–4.91 (*m*, 2H, *1-H*, *2-H*), 4.35–4.02 (*m*, 3H, *3-H*, *4-H*, *5-H*), 3.70–3.46 (*m*, 2H, *6-H*), 2.36–2.10 (*m*, 2H, *1″-H*), 1.95–1.85 (*m*, 3H, *1′-H*), 1.15–0.00 (*m*, 3H, *2″-H*).

**Degree of esterification per ring** (determined by nuclear magnetic resonance spectroscopy): 2.2 acryl-, 0.85 butyl-, 0.2 acetyl groups.

#### 2.3.8. Synthesis of Phosphorylated Cellulose Acrylate Propionate with PAHE (**6-CeAcPr-PAHE**)

A desiccated 100 mL Schlenk flask was filled with 2 g of **5-PAHE** in 8 mL anhydrous acetonitrile. The gas phase was flushed again with nitrogen, before 1 g of the **4-CeAcPr** was added with 10 mg of phenothiazine. An amount of 1.3 mL triethylamine in 12 mL anhydrous acetonitrile was added through a septum via a syringe. The oil bath was set to 70 °C under severe stirring in a nitrogen atmosphere.

The PAHE solution was added portion-wise within 40 min through the septum. When the addition was complete, the temperature was raised to 78 °C and the solution was stirred for 4.5 h. After cooling to room temperature, the easily volatile reagents/solvents were captured in a cooling trap, 25 mL of anhydrous toluene was added, and the temperature was raised to 50 °C. A brown solid precipitated and the solution was decanted. The precipitate was dissolved again in 4 mL anhydrous acetonitrile and precipitated by addition of 25 mL toluene. This procedure was repeated one more time to purify the product. Finally, the product was dried at 90 °C and then the temperature was continuously increased to 155 °C in vacuum (1 × 10^−2^ mbar) for 4 h, and 1.35 g was collected as a hygroscopic white powder. Yield: 75%.

**^1^H NMR** (300 MHz, d_6_-DMSO, ppm): δ 5.23–5.13 (*m*, 2H, *trans-6‴-H*, *trans-8‴-H*), 4.63, 4.27, 3.76, 3.37 (*m*, Cellulose, *not assignable*), 3.97 (*m*, 2H, *5‴-H*, *9‴-H*), 3.49 (*m*, 2H, *cis-6‴-H*, *cis-8‴-H*), 2.73–2.19 (*m*, overlap of: *1′-H*, *1″-H*, *2‴-H*, *3‴-H*), 1.06–0.96 (*m*, 3H, *2″-H*).

**^31^P NMR** (300 MHz, d_6_-DMSO, ppm): δ 50.51 (*s*, 1P).

#### 2.3.9. Synthesis of Phosphorylated Cellulose Acrylate Propionate with PAHE (**6-CeAcPr^SA^-PAHE**)

A desiccated 250 mL Schlenk flask was filled with 9 g of **4-CeAcPr^SA^** and 50 mg phenothiazine and dissolved in 75 mL anhydrous acetonitrile at 60 °C under a nitrogen atmosphere. An amount of 12 mL triethylamine was added through a septum via a syringe and the oil bath temperature was set to 70 °C. A solution of 9.2 g **5-PAHE** in 36 mL anhydrous acetonitrile was added portion-wise within 40 min through the septum. When the addition was complete, the temperature was raised to 78 °C and the solution was stirred for 4 h. After cooling to room temperature, the easily volatile reagents/solvents were captured in a cooling trap, 225 mL of anhydrous toluene was added, and the temperature was raised to 50 °C. A brown solid precipitated and the solution was decanted. The precipitate was dissolved again in 35 mL anhydrous acetonitrile, precipitated by addition of 125 mL toluene, and decanted again. This procedure was repeated three more times to purify the product. Finally, the product was dried at 100 °C and then continuously increased to 135 °C in vacuum (1 × 10^−2^ mbar) for 5 h, and 13.5 g was collected as a hygroscopic white powder. Yield: 78%.

**^1^H NMR** (300 MHz, d_6_-DMSO, ppm): δ 5.26–5.13 (*m*, 2H, *trans-6‴-H*, *trans-8‴-H*), 4.63, 4.27, 3.77, 3.20 (*m*, Cellulose, *not assignable*), 3.97 (*m*, 2H, *5‴-H*, *9‴-H*), 3.49 (*m*, 2H, *cis-6‴-H*, *cis-8‴-H*), 2.73–1.90 (*m*, overlap of: *1′-H*, *1″-H*, *2‴-H*, *3‴-H*), 1.06–0.96 (*m*, 3H, *2″-H*).

**^31^P NMR** (300 MHz, d_6_-DMSO, ppm): δ 50.51 (*s*, 1P).

#### 2.3.10. Synthesis of Phosphorylated Cellulose Acrylate Propionate with DOPO (**6-CeAcPr-DOPO**)

A 500 mL three-neck Schlenk round flask was equipped with a KPG stirrer and desiccated three times. The flask was filled with 8.7 g of the **4-CeAcPr**, 100 mg phenothiazine, 120 mL anhydrous acetonitrile, and 12 mL triethylamine. The solution was stirred heavily at 70 °C. An amount of 12 g of **5-DOPO** was dissolved in anhydrous acetonitrile and added within 45 min to the solution over a syringe through a septum at 75 °C. The product precipitated during the reaction and was isolated by filtration. The precipitate was purified by dissolving it in anhydrous acetonitrile at 135 °C, followed by reprecipitation from toluene and decantation. The product was dried under vacuum and 12 g of the flame retardant was collected. Yield: 64%.

**^1^H NMR** (300 MHz, CDCl_3_, ppm): δ 7.82–7.15 (*m*, 8H, *aromatic DOPO-H*), 4.96–4.73 (*m*, 2H, *1-H*, *2-H*), 4.31–3.57 (*m*, 5H, *3-H*, *4-H*, *5-H*, *6-H*), 2.51–2.22 (*m*, overlap of: *1′-H*, *1″-H*, *2‴-H*, *3‴-H*), 0.91 (*m*, 3H, *2″-H*).

**^31^P NMR** (300 MHz, CDCl_3_, ppm): δ 35.75 (*s*, 1P, main product: *DOPO ring closed*).

### 2.4. Preparation of Polymer Compounds and UL 94 V Test Specimens

All polypropylene polyethylene random copolymer (PP-*co*-PE) flame retardant compounds were prepared on a 12 mm co-rotating twin-screw extruder (ThreeTec, Seon, Switzerland) applying a mass temperature of 170 °C and a screw rotation speed of 50 rpm. The polyamide flame retardant compounds were prepared with the same twin-screw extruder and screw rotation speed but applying a mass temperature of 210 °C.

UL-94 test bars were injection-molded using a Haake Mini Jet Pro (Thermo Fisher Scientific, Karlsruhe, Germany). The PP-*co*-PE test specimens were prepared with a cylinder temperature of 200 °C and a mold temperature of 70 °C. To inject the polymer melt into the mold, a pressure of 700 bar was applied. The polyamide test specimens were prepared with a cylinder temperature of 235 °C, a tool temperature of 90 °C, and an injection pressure of 800 bar.

### 2.5. Flame Test: UL 94 Vertical/DIN EN 60695-11-10

The UL 94 flame tests were performed with test specimens of the dimensions 120 mm × 12.7 mm × 1.6 mm. The test specimens were conditioned for 48 h at 23 °C and 50% relative humidity before use.

## 3. Results and Discussion

To date, there has been no report of a synthesis route for cellulose acrylates, which is feasible on larger scales with a high biobased content. We have made it our goal to develop a novel synthesis route for mixed esters of cellulose to increase not only the solubility but also the compatibility with a polymeric matrix. For this purpose, we attached (additionally to the acrylic acid groups) saturated carbonic acid groups as illustrated in Figure 1I. After esterification (**4**), the available acryl groups are accessible to further post-functionalization. Double bonds tend to react as nucleophiles, but because the double bond of the acrylic compound is in conjugation with the carbonyl group, it can react as an electrophile in a Phospha-Michael addition. Such a reaction is likely for esters in the presence of suitable and soft (according to the HSAB theory [47]) Michael donors, like phosphorous-based nucleophiles (**5**). Such a Phospha-Michael addition was performed with DOPO and PAHE as Michael donors (Figure 1II). Depending on the composition of the reaction mixture, cellulose esters with a tailor-made structure are accessible, i.e., the ratio of saturated and unsaturated ester groups is adjustable. The saturated aliphatic groups increase the solubility of the esters and FRs, hence improving the compatibility of the FRs with the polymeric matrix. The novel FRs as well as the cellulose esters were characterized by NMR spectroscopy (^1^H, ^13^C, ^31^P, COSY, and HSQC), SEC, FT-IR and UV-VIS. Their thermal properties were evaluated by TGA. Especially ^13^C NMR spectroscopy, particularly the signal integration, worked surprisingly well in contrast to the literature [48]. The correlation spectroscopy (COSY) and heteronuclear single quantum coherence spectroscopy (HSQC) data were helpful to assign the signals properly. The spectra are shown in the supporting information. Furthermore, a list of typical solvents is attached to the supporting information section with regards to the solubility of the novel components.

Two selected phosphorylated cellulose esters were subjected to elemental analysis (Section 2.3.4 and Section 2.3.5). Unfortunately, the values of carbon and phosphorus content were found to be somewhat lower, as to be expected by evaluation of NMR spectra. However, it should be taken into account that the information on the substitution degree provided by NMR is relatively imprecise. Therefore, the elemental composition can only be roughly estimated on the basis of NMR spectroscopy. Additionally, the deviations can be caused by the rather low overall phosphorus content in the flame retardants, which is close to the detection limit of the method.

### 3.1. Characterization of the Acrylic Acid Anhydride Solution (***2***)

For step I in Figure 1, it is necessary to use a reactive acrylic acid derivate. Figure 4 shows the general procedure for the synthesis of such an anhydride solution, which contains acrylic acid anhydride and very small amounts of the mixed anhydride of acrylic and acetic acid.

NMR spectroscopy can be used to determine the relative amount of the mixed acid anhydride product (Figure A1). The vinyl protons of the one-sided acrylic anhydride (**2.3**) are overlaid by the acrylic anhydride main product (**2**) peaks at (6.54–6.48) ppm and (6.18–6.01) ppm. Fortunately, the singlet of the acetyl group is visible at 2.22 ppm and calculation leads to a relative amount of ~1.8% of the one-sided product. Moreover, there are some smaller amounts of free acrylic acid (**2.2**) and acetic anhydride (**2.1**) which were not outlined for the sake of clarity. Depending on the latter application of the cellulose esters, the degree of purity of the used anhydride mixture is variable because acetyl ester groups may be useful to increase the solubility of the product on one hand, but on the other hand, it could simplify hydrolysis of the corresponding ester. Experimental evaluation reveals that if acetyl groups are not wished in the final product, a content of <2% of the one-sided product (**2.3**) in the anhydride mixture is sufficient. If acetic acid groups are requested (or are at least not disturbing), a content of (7–8)% is fine. Such small amounts are enough because acetic anhydride is much more reactive than acrylic anhydride and is the most reactive acid/anhydride in the homologous series. Therefore, the reaction will surely lead in favor of acetic esters before the acrylic esters are attached.

### 3.2. Characterization of the Cellulose Esters (***4***)

Besides some accidentally introduced acetic acid groups in the cellulose esters, propionate or butyrate groups are more favorable to optimize the properties of the product and can be intentionally introduced by some equivalents of propionic/butyric acid anhydride. Such cellulose acrylate propionate (**4-CeAcPr**) or butyrate (**4-CeAcBu**) esters were produced and characterized successfully. In the following, the evaluation was carried out in detail for **4-CeAcBu** with the aid of Figure 5 and Figure 6. The **4-CeAcPr** was characterized in analogy and the spectra can be found in the Supporting Information (Appendix A).

Considering the ^1^H NMR spectrum in Figure 5A, we can assign and quantify the type and degree of esterification. A clear proof whether some free hydroxyl groups are still available could not be deduced. We calculated a degree of 2.1 acrylic ester (orange: 3H, 5.64–6.47 ppm), 0.8–0.9 butyric ester (green: 2H, 2.07–2.30 ppm; 2H, 1.48–1.65 ppm; 3H, 0.82–0.97 ppm) and 0.27 acetic ester units (grey: 3H, 1.81–1.91 ppm) per ring. The integrals were normalized in reference to the proton attached to the C1 cellulose carbon (blue), which is shifted the most into the low field at (5.11–4.89) ppm, due to an electron-poor environment. This will be discussed more in detail for Figure 6.

The integrable ^13^C NMR spectrum is shown in Figure 5B. In contrast to the literature proceedings [48], the spectrum was surprisingly well resolved, and the peak assignments were easy to evaluate for our cellulose derivates. For the sake of clarity, broad or prickly signals were merged and, therefore, averaged for assignable functional groups. Again, the spectrum was normalized in reference to the C1 cellulose carbon.

The acrylic double-bond carbons are clearly visible at 132.15 ppm for the electron-poorer R-**CH**=CH2 carbon and at 127.30 ppm for the slightly electron-richer R-CH=**CH2** carbon. Unfortunately, due to the broad CDCl_3_ and MTBE peaks, only the C1 and C6 carbon of the cellulose backbone are visible in the spectrum but at least the integrals match perfectly. The low-field carbonyl (brown) and high-field alkyl carbons (green and grey) of the ester groups are cut out separately in Figure 6. With great efforts, we were able to obtain a high resolution of the peaks to distinguish between the three possible points of attack (C2, C3, and C6 carbon) for the esterification.

The general assignment of the butyric, acrylic, and acetic ester groups is shown in Figure 6A (brown) for the carbonyls and in Figure 6B (green: butyl-, grey: acetyl-) for their alkyl groups. For example, three peaks are visible in the region 173–171 ppm. Two of them are very close and one peak is more shifted towards the low field. The differences are caused by the type of original OH group (primary or secondary) as well as the different chemical environment of the two secondary OH groups. The integrals also indicate an even distribution between the three peaks, or more precisely between the three carbonyl carbon atoms attached to C2, C3, and C6 of cellulose. It is noteworthy that the ester attached to C2 exhibits the poorest electron environment, because the neighboring C1 is the poorest electron carbon of the ring, leading to a higher electron pull upon C2 and consequently upon the attached carbonyl group at C2. This pattern is also visible for the alkyl groups of the ester in Figure 6B (green and grey). The splitting might not be so strong, but the integrals still indicate the same distribution/pattern. The ester attached to C6 is (most likely) the one with the richest electron environment because C6 is the only CH_2_ carbon of cellulose. Therefore, the electron pull is not as extended as for CH groups, and this is causing a shift towards high field for the corresponding esters attached to position C6. The electron pull upon the ester attached to C3 is somewhere between that of C2 and C6, causing an overlay with the ester positioned at C6.

Indeed, the catalyst may also have a large impact on the thermal stability of the product. For example, sulfuric acid might be a strong Brønsted acid to catalyze the esterification reaction, but at the same time, functionalization with sulfuric acid groups and degradation of the cellulose backbone cannot be avoided. The resulting cellulose esters visually show a shade of grey. However, white or colorless flame retardants are often required in many applications. Despite their efficiency, strong and degrading acids like sulfuric or trifluoromethanesulfonic acid are therefore not suitable catalysts. Several catalysts were examined according to their efficiency, degree of degradation, and thermal stability of the cellulose esters. Finally, the use of ZnCl_2_ seems to be the most promising candidate for this purpose. The thermal stability of the corresponding FRs are evaluated in Section 3.5.

### 3.3. Characterization of Tetrahydrofuro[3,4-d][1,3,2]dioxaphosphole 2-oxide (***5-PAHE***)

Step II in Figure 1 requires the primary synthesis of a *P-H* active species, which can act as a nucleophile in the course of a Phospha-Michael addition. Here, we like to introduce a biobased alternative based on sugar alcohols, compared to the commercially available and petrochemical-based product, e.g., DOPO. Figure 7 shows the synthesis of **5-PAHE** starting from erythritol (**5.1**, sugar alcohol). An ion exchanger (Dowex^TM^ 50WX8) favors cyclization (**5.2**) followed by a reaction of the same with dimethyl phosphite. The scheme also shows the most likely side reaction (**5.3**) triggered by hydrolysis.

The corresponding ^1^H NMR and ^31^P NMR spectra of PAHE are shown in Figure A2 and Figure A3. The spectra indicate that the synthesis yielded two stereoisomers with a diastereomeric excess of ~51%. Noteworthy for the ^1^H NMR spectrum is the very high P-H coupling constant of *J*_P-H_ = 723.0 Hz. This causes a wide splitting of the doublet along the x-axis, which is clearly visible for the isolated isomer in Figure A4. The ^31^P NMR spectrum allows for the differentiation between the two isomers of the product (Figure A3A) and PAHE can be tested under the reaction conditions, to see whether it is stable upon hydrolysis (Figure A3B). Finally, it can be deduced that the product is clean and moderately stable under the reaction conditions. An isolation of one stereoisomer is not necessary for the following Phospha-Michael addition because the experimental evaluation has shown that the stereochemistry does not limit the success of the reaction nor the efficiency of the final flame retardant.

### 3.4. Characterization of the Flame Retardant (Phosphorylated Cellulose Esters, ***6***)

Finally, in step II, the nucleophiles (**5-PAHE** or 5-DOPO) can be attached to the polymeric backbone of the cellulose ester by a Phospha-Michael addition, according to Figure 1. The ^1^H, ^31^P, and ^13^C NMR spectra for the final FR based on **4-CeAcBu** are evaluated in the following, while the ones for the **4-CeAcPr** ester are attached to the Supporting Information (Appendix A).

Unfortunately, the resolution of the ^13^C NMR spectrum in Figure 8 is not as good as the one for the cellulose ester. However, we can evaluate some important information. The double-bond peaks around 120–140 ppm, as in Figure 5B, vanished completely. Additionally, the intense PAHE peaks between 81.71 and 72.83 ppm appear as well as the newly formed CH_2_-CH_2_ unit. Due to the worse resolution and the addition of PAHE (reduction of double bond), the carbonyl signals seem to overlap at 171.43 ppm. The cellulose signals are hardly visible, either due to the overlap with the PAHE signals or due to a lack of intensity. From the high field of the spectrum, only the terminal butyric-(γ)-CH_3_ (13.22 ppm) and the newly formed propyl-(α)-CH_2_ (26.78 ppm) attached to the carbonyl can be distinguished. The other signals between 21.36 and 17.53 ppm can be assigned to the different CH_2_/CH_3_ groups of the acetyl-, (newly) propyl-, and butyl ester groups. The butyl-(α)-CH_2_ group attached to the carbonyl group is probably overshadowed by the DMSO solvent peak (see Figure 5B as comparison).

The ^31^P NMR spectrum in Figure 9A indicates the success of the reaction. Only one distinct peak at 50.45 ppm is detected, which means that the phosphacycle is still closed. The ^1^H NMR spectrum in Figure 9B shows the specific peaks of the PAHE group overlapping with the cellulose peaks. Therefore, integration is difficult. Still, the acrylate double-bond protons vanished, indicating a successful reaction. Following from this, the newly formed C-C single-bond protons should be visible in the spectrum, but they are overlapping with the acetyl-CH_3_, butyl-CH_2_, and d_6_-DMSO signals (2.08–2.72 ppm).

Altogether, the new synthetic route for the novel flame-retardant systems is feasible, and the collected products were tested regarding their thermal stability and flame retardancy. The results are shown and discussed in Section 3.5. For this purpose, DOPO-functionalized cellulose esters (**6-CeAcBu-DOPO**) were used as a reference. The related spectra of those flame retardants are also attached to the Supporting Information (Appendix A).

### 3.5. Thermal Analysis of the Flame Retardants

A selection of the synthesized biobased flame retardants on the basis of organic cotton (cellulose) with propionate and butyrate cellulose ester side groups and functionalized with PAHE or DOPO is shown in Table 1. Additionally, the used catalysts for the esterification reaction on the cellulose are given in Table 1 to clarify their potential influence on the thermal stability of the cellulose educts before functionalization. All three depicted flame retardants were analyzed concerning their thermal stabilities in TGA experiments under an inert gas atmosphere. The DOPO-functionalized flame retardant **6-CeAcBu-DOPO,** in contrast to its PAHE-functionalized counterparts, exhibits the highest thermal stability of all synthesized flame retardants (Figure 10A,B).

At a temperature of 295 °C, only 2% of mass loss occurs (see Table 2). In direct comparison, the PAHE-functionalized flame retardant **6-CeAcBu-PAHE**, with the only difference compared to **6-CeAcBu-DOPO** in its DOPO-based functional group while possessing the same cellulose-type modification and catalyst, is stable only up to 288 °C (2% mass loss). In general, all PAHE-functionalized flame retardants show a comparable thermal decomposition behavior (Figure 10). However, flame retardant **6-CeAcBu-PAHE**, shows a higher thermal stability (of about 21 °C) at 2% mass loss than flame retardant **6-CeAcPr^SA^-PAHE.** The difference here lies in the used catalyst for cellulose esterification: while **6-CeAcBu-PAHE** was synthesized with zinc chloride as a catalyst, **6-CeAcPr^SA^-PAHE** was synthesized with sulfuric acid (Table 2). In general, all flame retardants exhibit three main decomposition steps, but the exact temperature for each step differs for every single flame retardant. Nevertheless, all PAHE-functionalized flame retardants (**6-CeAcPr^SA^-PAHE and 6-CeAcBu-PAHE**) show rather similar temperature ranges for their first (278–308 °C), second (313–318 °C), and third (455–465 °C) decomposition steps (Figure 10B). In contrast, DOPO-functionalized flame retardant **6-CeAcBu-DOPO** decomposes at higher temperatures for the first (325 °C) and second (375 °C) decomposition steps, but step three occurs at lower temperatures (424 °C, Figure 10). It is noteworthy to mention that, when sulfuric acid is used as the catalyst for the cellulose esterification (PAHE-functionalized flame retardant **6-CeAcPr^SA^-PAHE**), the resulting flame retardant exhibits the lowest thermal stability of all analyzed flame retardants (first decomposition step at 278 °C, Figure 10B). Out of the group of PAHE-functionalized flame retardants, the most thermally stable one is **6-CeAcBu-PAHE**. It shows its first decomposition step at 308 °C, which is about 30 °C higher than for **6-CeAcPr^SA^-PAHE** catalyzed with sulfuric acid. Concerning the residual masses in the TGA experiments, a big difference between the DOPO- and the PAHE-functionalized flame retardants can be noted: DOPO-functionalized **6-CeAcBu-DOPO** leaves only about half the amount of residual mass (18.7%) than the PAHE-functionalized flame retardants (32.6–36.7%) in the TGA experiments (Table 2). PAHE-functionalized flame retardant **6-CeAcPr^SA^-PAHE** catalyzed with sulfuric acid gives the lowest residual mass (32.6%) of all flame retardants tested, whereas flame retardant **6-CeAcBu-PAHE**, which was synthesized with zinc chloride as the esterification catalyst, yields the highest residual mass (36.7%), which can be beneficial in terms of flame-retardant condensed-phase activity in a polymer compound.

It is noteworthy to mention that moving from organic cotton as an educt to PAHE-functionalized flame retardant **6-CeAcBu-PAHE**, the residual mass at 800 °C in TGA experiments under nitrogen could be increased from only 7.8% to 36.7% (Figure 11A), which may result in a potentially increased flame-retardant condensed-phase activity.

### 3.6. Processing and Thermal Characterization of Polymer Compounds

As depicted in Section 3.5, flame retardant **6-CeAcBu-PAHE** was used for the subsequent compounding experiments because it is the most promising derivative according to its thermal properties. Even though DOPO-functionalized flame retardant **6-CeAcBu-DOPO** exhibits a slightly higher thermal stability in TGA experiments (2% mass loss at 295 °C), it was not further considered in the compounding experiments because of its lack of sustainability in contrast to PAHE (**5**), which originates from erythritol (**5.1**), a renewable raw material. Additionally, the DOPO-based structure of **6-CeAcBu-DOPO** was not expected to work particularly well in PP-*co-*PE and PA, which was also supported by preliminary test results.

PAHE-functionalized flame retardant **6-CeAcBu-PAHE** was at first used for the compounding experiments in a random polypropylene-polyethylene copolymer (PP-*co*-PE*, Moplen^®^ RP320 M*, LyondellBasel), as those copolymer types are commonly used in a wide range of applications such as high-transparency packaging, blow molding, and injection molding. For the biobased polyamide (PA, *Ultramid^®^ Flex F29*, BASF SE), a flame-retardant solution is not yet developed as its primary use is in food packaging. However, due to its excellent properties such as improved transparency, low water absorption, and softness, there is a rising interest in rendering it flame retardant while maintaining its optical and mechanical properties. It was therefore used for the recipe development in polyamide with the PAHE-functionalized flame retardant **6-CeAcBu-PAHE**.

Figure 12 indicates the chosen compounding temperatures for the PP-co-PE as well as for the PA formulations, and additionally depicts the decomposition temperatures of flame retardant 6-CeAcBu-PAHE in comparison with the two base polymers. As both applied compounding temperatures are far below the critical decomposition temperature of the flame retardant (Figure 12A), especially in PP-co-PE, there were no major issues encountered in compounding and injection molding of the desired UL 94 V specimens. However, an excellent distribution of 6-CeAcBu-PAHE could not be achieved at loadings of 8–10 wt.% in PP-co-PE (Table 3) due to its rather polar nature. As a consequence, the prepared UL 94 test specimens appeared at least partially inhomogeneous on visual inspection (see micrographs in Figure A6). Table 3 summarizes the prepared formulations in PP-*co*-PE, where decabromodiphenyl ethane (DBDE) and antimony(III) oxide were used as the state-of the-art halogenated reference [24] (formulation II-PP-PE) and the organic disulfide poly-*tert.*-butylphenol disulfide (PBDS) as the sole flame retardant and reference at 10 wt.% loading (formulation III-PP-PE) and as the synergist [23] for flame retardant **6-CeAcBu-PAHE** at 2 wt.% loading (formulation V-PP-PE).

Flame retardant **6-CeAcBu-PAHE** was investigated at 10 wt.% loading alone (formulation IV-PP-PE) or at 8 wt.% loading in combination with PBDS (formulation V-PP-PE, see Table 3). Thermogravimetric analyses under nitrogen were performed for all prepared compounds and are depicted in Figure 13.

PP-*co*-PE copolymer shows a well-known one-step decomposition [49] with a maximum decomposition at 458 °C (Figure 13A,B, orange curves). When flame retardant **6-CeAcBu-PAHE** is added in compound IV-PP-PE, an additional decomposition step occurs at 302 °C, which corresponds to the main decomposition of **6-CeAcBu-PAHE** as shown in Figure 10 and Figure 11. The same decomposition step can also be found for compound V-PP-PE (see Table 3) containing 8 wt.% **6-CeAcBu-PAHE** and 2 wt.% PBDS but with the expected higher residual mass of approx. 92% at 350 °C, in contrast to 90% residual mass at 350 °C for compound IV-PP-PE containing 10 wt.% **6-CeAcBu-PAHE** (Figure 13A). On the other hand, both compounds only exhibit residual masses at 800 °C, which is far below the 10 and 8 wt.% dosing of **6-CeAcBu-PAHE** in both recipes, indicating a chemical reaction at decomposition which also involves the phosphorous functional groups of the FR. A striking effect occurs with compound III-PP-PE containing 10 wt.% PBDS, where the maximum decomposition temperature is shifted to 442 °C only while maintaining a single, one-step decomposition, indicating a chemical interaction between the polymer and PBDS and a significant destabilization of the polymer. Even more noteworthy is the decomposition behavior which can be observed with compound V-PP-PE containing 8 wt.% flame retardant **6-CeAcBu-PAHE** and 2 wt.% PBDS, where both the flame retardant and polymer decomposition steps can be detected at 301 °C and 463 °C (maximum mass losses), respectively, but an additional decomposition step occurs as a broad shoulder at about 380–400 °C.

This could be an indication of a chemical interaction either between flame retardant **6-CeAcBu-PAHE** and synergist PBDS or a reaction product of PBDS with the PP-*co*-PE copolymer, and this should be investigated in more detail.

Table 4 shows the prepared formulations in *Ultramid^®^ Flex F29*. PAHE-functionalized flame retardant **6-CeAcBu-PAHE** was investigated as a sole flame retardant at 20 wt.% loading (formulation III-PA) and compared to a halogen-free, 20 wt.% DEPAL formulation which was expected to render the translucent *Ultramid^®^ Flex F29* opaque, which is generally an undesired side effect. It has to be mentioned that, in contrast to DEPAL, the distinct hygroscopicity and residual water content of flame retardant **6-CeAcBu-PAHE** even after vacuum drying caused severe issues, not only at the compounding step but also during injection molding of the UL 94 test specimens. The obtained PA compound containing **6-CeAcBu-PAHE** at 20 wt.% loading (formulation III-PA) and the corresponding injection-molded UL 94 test bars prepared from compound III-PA were strongly discolored, and the desired translucency could not be obtained in these first tests either.

The thermogravimetric analysis of compound III-PA containing 20 wt.% flame retardant **6-CeAcBu-PAHE** is depicted in Figure 14 and compared to the base polymer only (formulation I-PA). It is very evident that, after compounding in the presence of **6-CeAcBu-PAHE** at 210 °C for 1.5 min, the decomposition temperature of the base PA is strongly reduced (2% of mass loss at 279 °C), and instead of a single-step decomposition at 457 °C, two additional decomposition steps at lower temperatures (408 °C and 362 °C, respectively) can be observed (Figure 14B). This indicates that during compounding with the hygroscopic flame retardant **6-CeAcBu-PAHE**, *Ultramid^®^ Flex F29* was unfortunately already significantly decomposed, which could also already be observed visually. The greatly reduced melt viscosity after compounding (and injection molding) also speaks for this rather undesired effect and has to be addressed in further studies.

### 3.7. Flame Retardant Results

For flame-retardant polymer formulations in electrical and electronic applications, the UL 94 vertical flame test is the globally valid safety standard to determine the flammability of plastic materials [50]. It was therefore used to investigate the flame-retardant potential of the synthesized flame retardants in the prepared polymer compounds. In a standard PP-*co*-PE copolymer formulation, PAHE-functionalized **6-CeAcBu-PAHE** was investigated as a sole flame retardant at 10 wt.% loading (formulation IV-PP-PE) or in combination with the organic disulfide PBDS at 8 wt.% loading (formulation V-PP-PE). DBDE and ATO were used as a halogenated reference at a total 10 wt.% loading (formulation II-PP-PE, see Table 3). The UL 94 V test was performed with 1.6 mm injection-molded specimens. The results are presented in Figure 15.

Formulation IV-PP-PE containing 10 wt.% flame retardant **6-CeAcBu-PAHE** shows rather short burning times after the first ignition but fails to reach a UL 94 classification due to relatively long burning times > 50 s after the second ignition, combined with burning dripping (Figure 15). Synergist PBDS alone at equal loading (formulation III-PP-PE, see Table 3) equally does not reach a UL 94 classification but already exceeds > 50 s of burning at the first ignition for five out of five tested specimens. Only when flame retardant **6-CeAcBu-PAHE** (8 wt.%) is combined with PBDS (2 wt.%) in formulation V-PP-PE does a synergetic interaction occur, leading to very short burning times << 10 s after the first and second ignitions for all five out of five tested specimens. Regardless of the pleasingly short afterburning times, the tested specimens exhibited a strong tendency to release burning drips which ignited the cotton underneath, leading to an overall UL 94 V-2 classification only. These findings correlate well with the TGA results obtained for formulation V-PP-PE (Figure 13), which already indicated an influence on the PP-*co*-PE copolymer degradation and a potential interaction of the flame retardant and synergist, leading to an additional decomposition step at approximately 385 °C (Figure 13B) and a significantly improved UL 94 performance (Figure 15).

The detailed UL 94 test results for the investigated PP-*co*-PE formulations are shown in Table A1 in Appendix C.

In PA, PAHE-functionalized flame retardant **6-CeAcBu-PAHE** was investigated as a sole flame retardant at 20 wt.% loading (formulation III-PA) and compared to a halogen-free, 20 wt.% DEPAL formulation (Figure 16). Despite the already mentioned effect that all obtained UL 94 test bars of formulation III-PA were strongly discolored after injection molding, all five tested 1.6 mm specimens exhibited very short afterburning times of ≤2 s, and a second ignition did not occur (Figure 16). Only one out of five tested specimens generated burning drips which ignited the cotton, yielding an overall UL 94 V-2 classification for formulation III-PA (Table 4). However, it has to be noted that, in direct comparison to formulation II-PA containing DEPAL at 20 wt.% loading, flame retardant **6-CeAcBu-PAHE** in formulation III-PA exhibited the by far shorter overall burning times at equal loading (Figure 16).

In summary, it can be concluded that despite the discussed issues caused by flame retardant **6-CeAcBu-PAHE**’s strong hygroscopicity, the overall UL 94 V performance and especially the very short first in combination with the non-existent second afterburning times proved to be promising towards the development of a profound UL 94 V-0 classification with PAHE-functionalized flame retardants for biobased polyamides. The detailed UL 94 test results for the investigated PA formulations are presented in Table A2 in Appendix C.

## 4. Conclusions

In summary, a novel set of biobased FRs based on cellulose and sugar alcohols was successfully synthesized, characterized, and its flame-retardant performance was evaluated in a random polypropylene-polyethylene copolymer and in an aliphatic and partially biobased polyamide. The novel cellulose-ester flame retardants were especially characterized by ^13^C-NMR spectroscopy (300 MHz). Particularly, the integration of the cellulose ester signals and the success rating of the followed Phospha-Michael addition worked surprisingly well. In the past years, sugar alcohols such as erythritol became widespread in the food industry and hence produced in large quantities from biobased sources. Besides that, the optimized procedure for the in-house made acrylic anhydride solution and the esterification procedure are closed-loop processes and free from disposable waste products. Furthermore, the novel esterification approach is also not tied to toxic or halogenated reactants and is quite similar to the industrial esterification procedure of cellulose triacetate. Therefore, the synthesized components in this work might be of high value not only from an academic but also from an industrial point of view. In a series of orientating UL 94 V flame experiments in PP-*co*-PE copolymer, the PAHE-modified butyrate ester flame retardant **6-CeAcBu-PAHE** showed remarkably short burning times at 8 wt.% loading when combined with PBDS as a synergist but at the same time exhibited burning dripping which led to an overall UL 94 V-2 classification only. When used in aliphatic PA at 20 wt.% loading without any additional synergist, the resulting specimens resulted in a UL 94 V-2 classification as well, with only sporadic burning dripping at very short overall burning times.

However, there is some room for improvement. For example, it is highly desirable to move away from cotton and use wood or straw cellulose instead, which often accrue as still largely unused byproducts from agriculture and forestry. Derivatives from sugar alcohols, such as xylitol or mannitol, might also be interesting alternatives to enhance the thermal stability of the FRs further. The high hygroscopicity of the synthesized flame retardants caused some detectable polymer decomposition in PA during compounding and injection molding, which will be addressed in future works as it is a crucial step towards the development of viable and commercially attractive flame-retardant solutions. These promising test results will be the base for more detailed investigations and structural optimizations of the novel FRs in future.

## 5. Patents

Michael Ciesielski, Elias Chalwatzis, Robin Nezami, WO 2021/048335, 2021 [19].

## Figures and Tables

**Figure 1 polymers-15-03195-f001:**
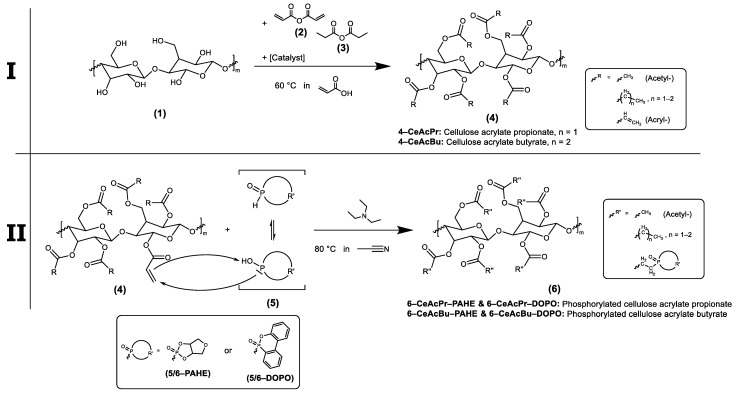
Schematic representation of the (**I**) cellulose esterification (**1** → **4**) followed by the (**II**) Phospha-Michael addition with DOPO or PAHE (**4** → **6**).

**Figure 2 polymers-15-03195-f002:**
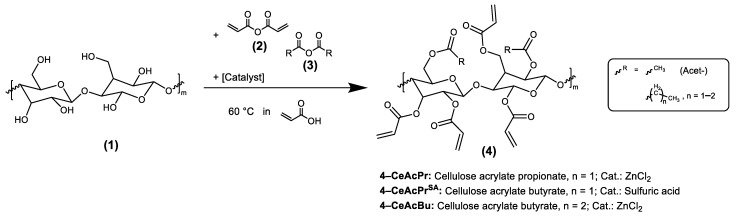
Synthesis procedure of the cellulose esterification.

**Figure 3 polymers-15-03195-f003:**
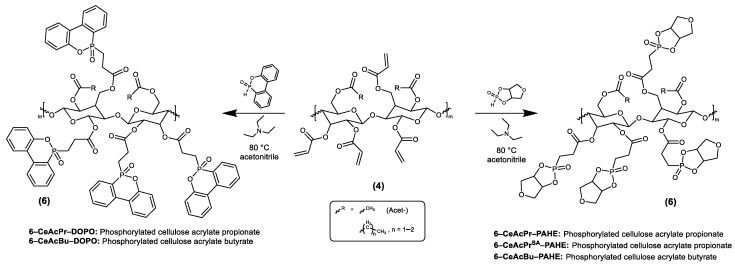
Synthesis procedure of the Phospha-Michael addition with 6H-Dibenzo[c,e][1,2]oxaphosphorin-6-on (DOPO) and phosphorylated anhydro erythritol (PAHE).

**Figure 4 polymers-15-03195-f004:**
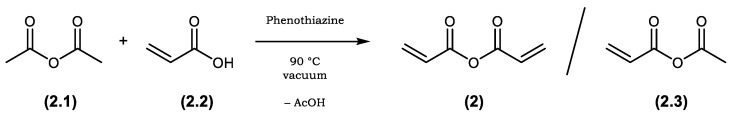
Reaction scheme for the synthesis of the anhydride mixture (**2**, **2.3**) from acetic anhydride (**2.1**) and acrylic acid (**2.2**).

**Figure 5 polymers-15-03195-f005:**
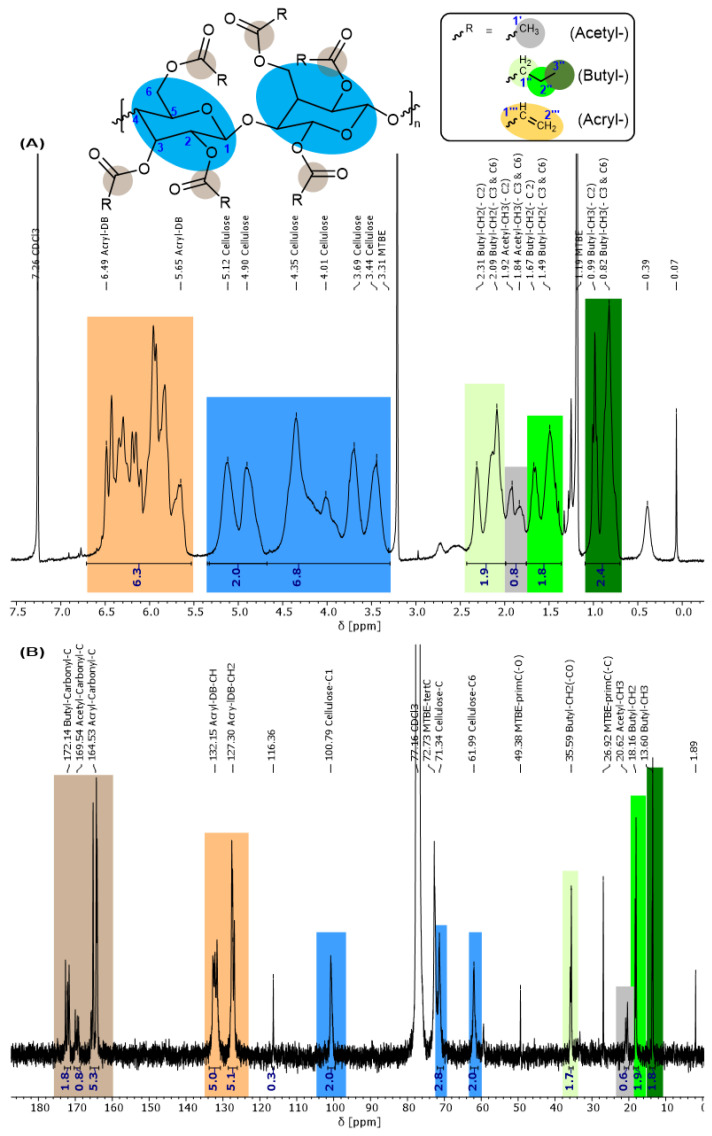
The functional groups are highlighted in colors according to their peak. (**A**) ^1^H NMR spectrum of cellulose acrylate butyrate (**4-CeAcBu**) in CDCl_3_ (300 MHz). Chloroform: 7.26 ppm (and 0.05 ppm impurity), MTBE: 3.20 ppm and 1.17 ppm. (**B**) ^13^C NMR spectrum of cellulose acrylate butyrate (**4-CeAcBu**) in CDCl_3_ (300 MHz). Only C1 and C6 of the Cellulose backbone (blue) are clearly visible due to an overlay with the solvent peaks (CDCl_3_ and MTBE). Chloroform: 77.16 ppm (and 1.89 ppm impurity), MTBE: (26.92, 49.38, and 72.73) ppm.

**Figure 6 polymers-15-03195-f006:**
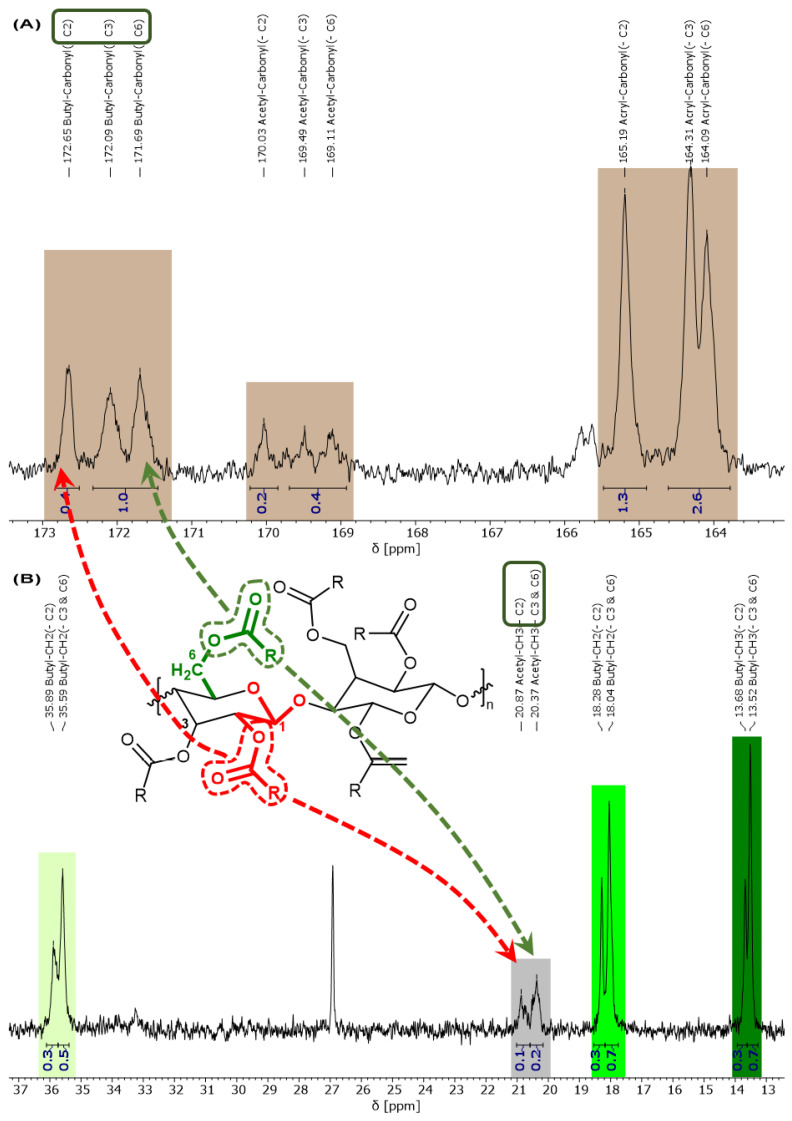
^13^C NMR spectrum cutout (**A**) 163–173 ppm and (**B**) 13–37 ppm of cellulose acrylate butyrate **4-CeAcBu** in CDCl**_3_** (300 MHz). The functional groups are highlighted in colors according to their peak. The splitting pattern and integrals indicate that a distinction between the C2, C3, and C6 carbon of the cellulose backbone is possible and was illustrated with green (electron-richer ester) and red (electron-poorer ester) arrows. MTBE: 26.92 ppm.

**Figure 7 polymers-15-03195-f007:**
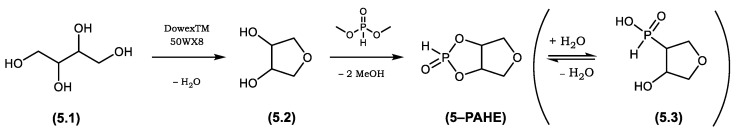
Synthesis of **5-PAHE** performed by a catalyzed cyclisation followed by a reaction with dimethyl phosphonate.

**Figure 8 polymers-15-03195-f008:**
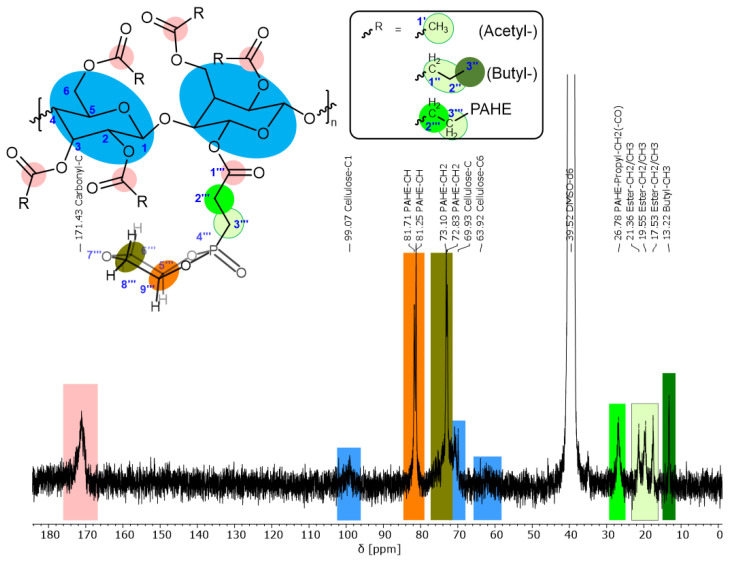
^13^C NMR spectra of the phosphorylated cellulose acrylate butyrate (**6-CeAcBu-PAHE**) in d_6_-DMSO (300 MHz). The double-bond signal at (120–140) ppm vanished, indicating the success of the Phospha-Michael addition.

**Figure 9 polymers-15-03195-f009:**
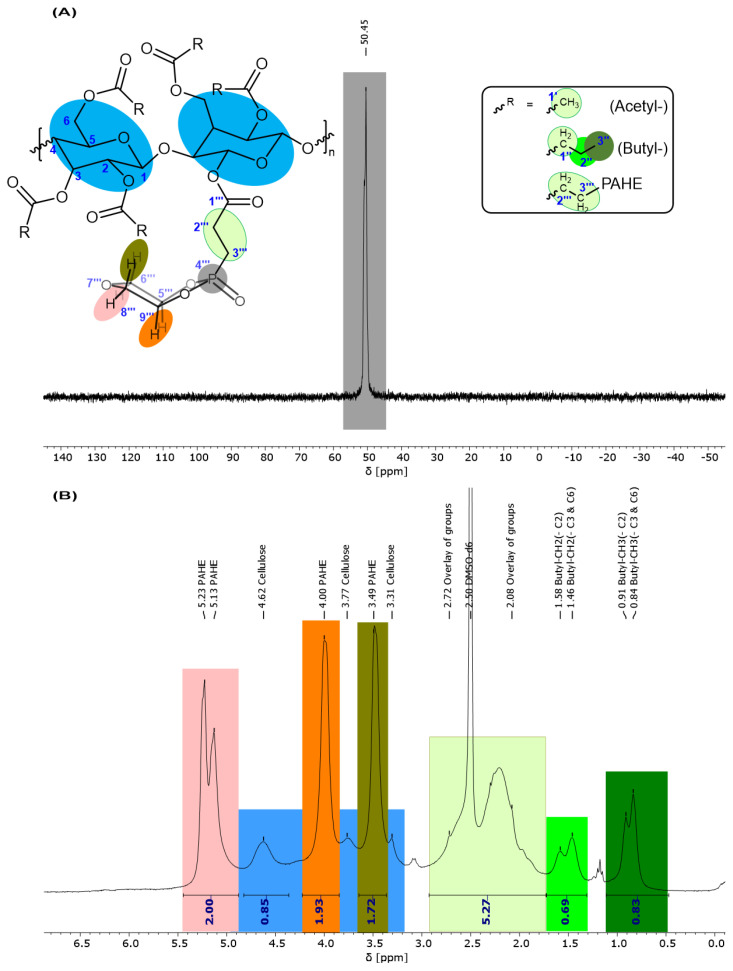
(**A**) ^31^P NMR spectra of the final phosphorylated cellulose acrylate butyrate (**4-CeAcBu-PAHE**) in d_6_-DMSO (300 MHz). (**B**) ^1^H NMR spectrum of the same (300 MHz). The functional groups are highlighted in colors according to their peak. The PAHE signals are overlapping with the cellulose signals as well as the overlay marked at 2.21 ppm (lime green). The latter contains acetyl-CH_3_, butyl-CH_2_, and the newly formed -CH_2_-CH_2_ group from the former acrylic double-bond -CH=CH_2_. The spectrum still shows some impurities at 1.18 ppm and 3.09 ppm.

**Figure 10 polymers-15-03195-f010:**
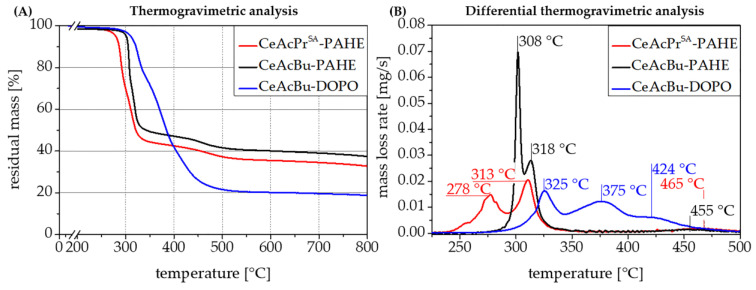
(**A**) Thermogravimetric analyses of the synthesized flame retardants (Table 1). (**B**) Corresponding mass loss rates from differential thermogravimetric analyses (DTGs). Analyses were performed under N_2_ atmosphere with a heating rate of 10 K/min.

**Figure 11 polymers-15-03195-f011:**
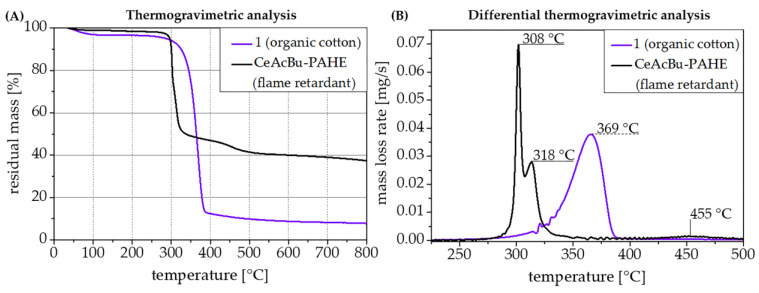
(**A**) Thermogravimetric analysis of organic cotton **1** compared to flame retardant **6-CeAcBu-PAHE**. (**B**) Differential thermogravimetric analyses (DTGs). Analyses were measured under N_2_ atmosphere with a heating rate of 10 K/min.

**Figure 12 polymers-15-03195-f012:**
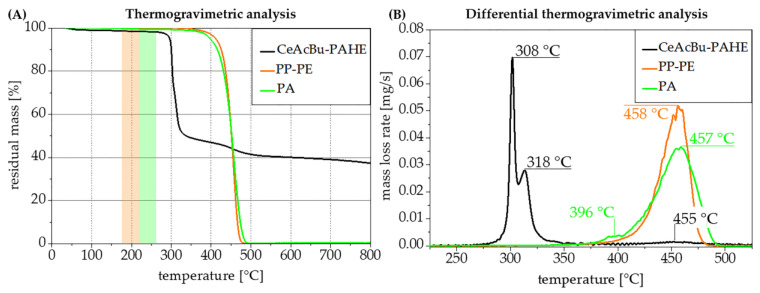
(**A**) Thermogravimetric analyses of flame retardants **6-CeAcBu-PAHE** (black) in comparison with the polymers used for compounding. Orange (PP-co-PE Moplen^®^ RP320 M) and green (PA Ultramid^®^ Flex F29). (**B**) Differential thermogravimetric analysis. Analyses were performed under a N_2_ atmosphere with a heating rate of 10 K/min.

**Figure 13 polymers-15-03195-f013:**
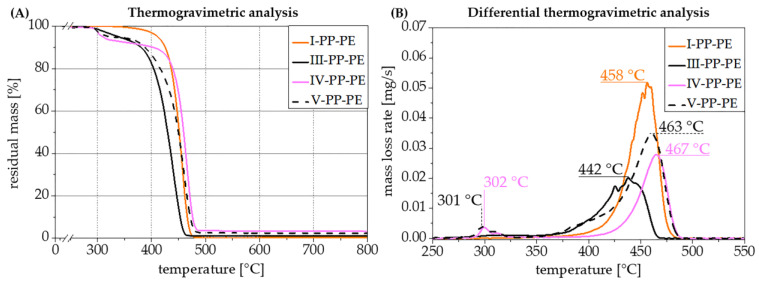
(**A**) Thermogravimetric analyses of the prepared flame-retardant formulations in PP-co-PE. (**B**) The corresponding differential thermogravimetric analyses (DTGs). Analyses were performed under a N_2_ atmosphere with a heating rate of 10 K/min.

**Figure 14 polymers-15-03195-f014:**
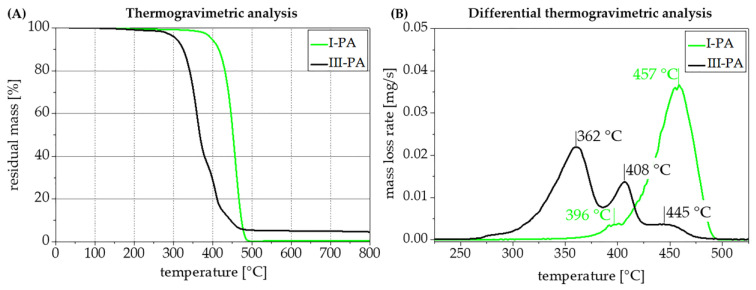
(**A**) Thermogravimetric analyses of formulation I-PA and III-PA measured under N_2_ atmosphere with a heating rate of 10 K/min and (**B**) the corresponding mass loss rates (DTGs).

**Figure 15 polymers-15-03195-f015:**
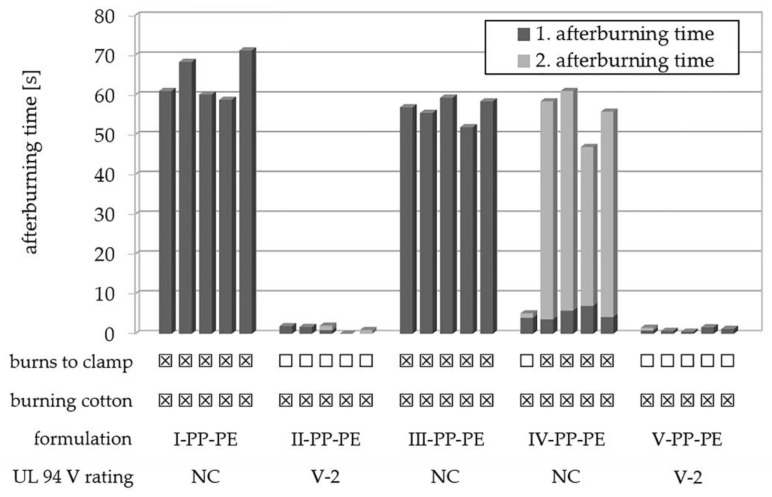
UL 94 V (1.6 mm) test results of flame retardant 6-CeAcBu-PAHE alone and in combination with PBDS as synergist in PP-co-PE. DBDE and ATO are used as halogenated references at equal total loading (formulations see Table 3). NC = non-classified, V-2 (rating) = cotton ignition, single burning time < 30.

**Figure 16 polymers-15-03195-f016:**
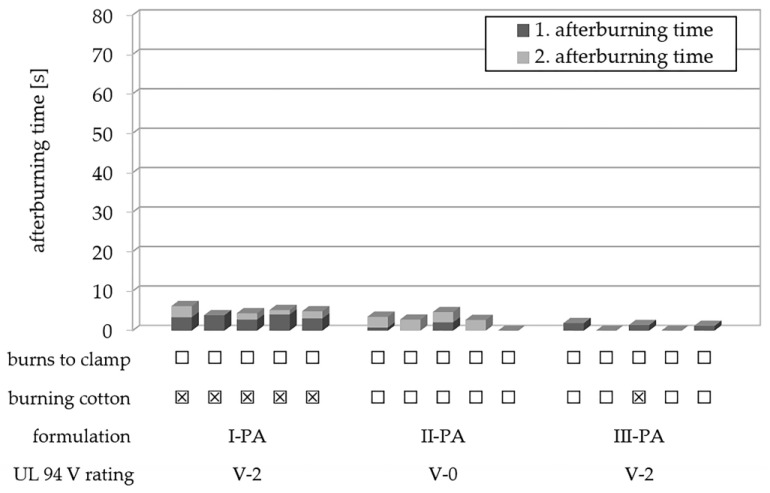
UL 94 V (1.6 mm) test results of flame retardant 6-CeAcBu-PAHE in PA. DEPAL (20 wt.% in PA) is shown as a halogen-free reference. V-2 (rating) = cotton ignition, single burning time < 30.

**Table 1 polymers-15-03195-t001:** Overview of synthesized flame retardants: educts, catalyst used in synthesis, and chemical functionalization.

Flame Retardant	Cellulose Type	Cellulose Modification	Catalyst for Esterification	Functional Group
**6-CeAcPr^SA^-PAHE**	cotton	propionate ester	H_2_SO_4_	PAHE
**6-CeAcBu-PAHE**	cotton	butyrate ester	ZnCl_2_	PAHE
**6-CeAcBu-DOPO**	cotton	butyrate ester	ZnCl_2_	DOPO

**Table 2 polymers-15-03195-t002:** Thermal stabilities and residual masses of the synthesized flame retardants (see also Figure 10).

Flame Retardants	2% Mass Loss [°C]	5% Mass Loss [°C]	Residual Mass at 800 °C [%]
**6-CeAcPr^SA^-PAHE**	267	277	32.6
**6-CeAcBu-PAHE**	288	297	36.7
**6-CeAcBu-DOPO**	295	312	18.7

**Table 3 polymers-15-03195-t003:** Prepared PP-co-PE formulations. Amounts of flame retardants are given in weight percent of total weight.

Formulation No.	Composition	6-CeAcBu-PAHE	PBDS	DBDE	ATO
I-PP-PE	PP-*co*-PE	-	-	-	
II-PP-PE	PP-*co*-PE +DBDE + ATO	-	-	6.7	3.3
III-PP-PE	PP-*co*-PE + PBDS	-	10	-	
IV-PP-PE	PP-*co*-PE + 6-CeAcBu-PAHE	10	-	-	
V-PP-PE	PP-*co*-PE + 6-CeAcBu-PAHE + PBDS	8	2	-	

**Table 4 polymers-15-03195-t004:** Prepared polyamide formulations. Amounts of flame retardants are given in weight percent of total weight.

Formulation No.	Composition	6.1B	DEPAL
I-PA	PA	-	-
II-PA	PA + DEPAL	-	20
III-PA	PA + 6-CeAcBu-PAHE	20	-

## Data Availability

Not applicable.

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
