# Peer review of "Novel Macromolecular and Biobased Flame Retardants Based on Cellulose Esters and Phosphorylated Sugar Alcohols"

_polymers, 2023, doi:10.3390/polym15153195_

Round 1
Reviewer 1 Report
The paper describes the synthesis of new macromolecular bio-based flame retardants based on cellulose esters and phosphorylated sugar alcohols. The synthesis procedures are well described and the new products are well characterized by Nuclear Magnetic Resonance Spectroscopy (NMR). The applications tested by the authors are encouraging
I point out a few typos:
- A mistake in the caption of Fig. 10 where the colours of the systems are inverted with respect to what indicated in the Figure insert
- A mistake on the first line of table 3 where PDBS is reported instead of PBDS
Author Response
Thank you for your well-disposed evaluation of our study!
We removed the errors as follows:
- A mistake in the caption of Fig. 10 where the colours of the systems are inverted with respect to what indicated in the Figure insert
Thank you very much for this note: We corrected the color set in Fig. 10.
- A mistake on the first line of table 3 where PDBS is reported instead of PBDS
Corrected.
Reviewer 2 Report
The authors report the modification of cellulose with different functional groups in order to obtain flame retardants. Likewise, they evaluate these substances in polymeric systems using melt mixing. It is an interesting work because they report the synthesis of these flame retardants modifying cellulose. However, it is necessary to improve the manuscript to make it easier to read and understand. For example:
1. Perform 2D NMR spectra, e.g., COSY, for a correct assignment of the peaks. This result can be placed in the supplementary material.
2. Mention in NMR methods whether the measurement was made in solid or solution.
3. Make a general figure that observes the routes used to obtain the different flame retardants and place in the section of materials and methods.
4. It is also necessary to place the infrared and UV-Vis spectra (material supplementary) of the products due to their few reports in the literature.
5. Report the physicochemical properties of the different products obtained (flame retardants), solubility, pH, color, among others.
6. Review the methods of synthesis of flame retardants, in some cases, it lacks reagents, and in others, steps are repeated. This section is confusing. You can also place here the reactions described in the results discussion section.
7. The SEC method lacks the solvent used in the sample, concentration and dn/dc of the samples.
8. In the discussion section correlating NMR and elemental analysis
9. The results of the thermal analysis are observed elevated concentrations of catalysts such as Zn and these may have an effect on the application being described. If this catalyst is removed, does it have the same effect as a flame retardant?
10. I also suggest looking at micrographs of the polymer and flame retardant mixture in order to see if they are miscible systems. They can also be supported with DSC and rheology.

Please see attachment for some grammatical errors.
It is necessary to check the manuscript grammatically.
Author Response
Thank you for thorough checking our manuscript!!
We performed following alterations and supplementations according to reviewer request:
- Perform 2D NMR spectra, e.g., COSY, for a correct assignment of the peaks. This result can be placed in the supplementary material.
Thank You for this valuable remark! Of course, 2D NMR spectra may support the peak assignment. Therefore, COSY as well as HSQC measurements were carried out . The COSY and HSQC spectra confirm the assignment of the peaks. These 2D spectra have been inserted into the supplementary information part.
- Mention in NMR methods whether the measurement was made in solid or solution.
Thank You for this note! The NMR spectra were recorded in solution. An information on the execution of liquid nuclear magnetic resonance spectroscopy (NMR) measurements with the use of deuterated chloroform and deuterated dimethyl sulfoxide as solvents is inserted into the methods chapter.
- Make a general figure that observes the routes used to obtain the different flame retardants and place in the section of materials and methods.
We willingly accepted this suggestion and introduced a scheme that illustrates all synthetic routes as well as the structures of the substances. We placed this scheme at the beginning of chapter 2.3 (Synthesis of flame retardants), i.e., directly after the section Materials and Methods.
- It is also necessary to place the infrared and UV-Vis spectra (material supplementary) of the products due to their few reports in the literature.
The reviewer is right: Novel substances should be characterized thoroughly including IR and UV Vis spectra. To complete the spectroscopic characterization these spectra were recorded and inserted into the supplementary information part. In case of IR spectra, wave numbers of selected absorption bands were inserted into the Synthesis part (2.3).
- Report the physicochemical properties of the different products obtained (flame retardants), solubility, pH, color, among others.
It is mentioned that the products were isolated as white powers in case of using zinc chloride as catalyst. When sulfuric acid was applied as catalyst, the substances showed a pale grey color. Moreover, it is noted that the novel flame retardants are hygroscopic. According to the reviewer request, information on the solubility of the obtained cellulose derivatives and flame retardants, respectively, are provided (supplementary information). The substances are not soluble in water. Therefore, pH determinations could not be conducted.
- Review the methods of synthesis of flame retardants, in some cases, it lacks reagents, and in others, steps are repeated. This section is confusing. You can also place here the reactions described in the results discussion section.
Thank you for thoroughly examining the Synthesis part! In fact, we identified a few errors and some mistakeable sentences and corrected them.
- The SEC method lacks the solvent used in the sample, concentration and dn/dc of the samples.
The following sentence has been inserted into the Methods chapter: “The sample was dissolved in chloroform (2 mg/mL) and was filtered with a 0.45 µm PTFE syringe filter prior to injection.”
- In the discussion section correlating NMR and elemental analysis
Thank you for this remark. In fact, there is a difference between the expectancy values estimated from NMR spectroscopy and the results of elemental analysis. We introduced following paragraph into the discussion section:
“Two selected phosphorylated cellulose esters were subjected to elemental analysis (chapter 2.3.4 and 2.3.5). Unfortunately, the values of carbon and phosphorus content were found to be somewhat lower as to be expected by evaluation of NMR spectra. However, it should be taken into account that the information on the substitution degree provided by NMR is relatively imprecise. Therefore, the elemental composition can only be roughly estimated on the basis of NMR spectroscopy. Additionally, the deviations can be caused by the rather low overall phosphorus contents in the flame retardants, which is close to the detection limit of the method.”
- The results of the thermal analysis are observed elevated concentrations of catalysts such as Zn and these may have an effect on the application being described. If this catalyst is removed, does it have the same effect as a flame retardant?
Thank You for this comment! In the current state of our investigations, it is not possible to answer this interesting question definitively. We intend to investigate the influence of catalyst traces on the thermal and flame-retardant properties of the novel additives in near future. Moreover, ongoing trials aim to diminish the amount of zinc catalyst necessary to conduct the cellulose esterification. We suppose that a pronounced impact of catalyst traces on the flame-retardant effect is rather unlikely.
- I also suggest looking at micrographs of the polymer and flame retardant mixture in order to see if they are miscible systems. They can also be supported with DSC and rheology.
Thank you very much for this comment! We added respective micrographs of the flame retardant compounds to the Appendix part.
It is necessary to check the manuscript grammatically.
We checked the manuscript carefully and removed a couple of grammar and spelling errors.
Reviewer 3 Report
The authors presented a set of macromolecular and biobased flame-retardants based on cellulose esters and phosphorylated sugar alcohols, which meet those challenging new sustainability requirements. The current level of the paper can meet the needs of POLYMERS.
Author Response
Thank you for your well-disposed evaluation of our study!
Round 2
Reviewer 2 Report
The authors made all suggested changes. Only minor errors were found in the supplemental material, specifically FTIR and UV-Vis. In the FTIR Figure place wavenumber (cm-1) and no value on the y-axis by this normalized. In the UV-Vis, Y axis use decimal point do not eat. The manuscript is well written, therefore I recommend its publication.
Author Response
We corrected the figures of the FTIR and UV-Vis spectra according to reviewer request.